# Threshold Moving for Online Class Imbalance Learning with Dynamic Evolutionary Cost Vector

**Peijia Qin**                                                    *qinpj2021@mail.sustech.edu.cn*
*Department of Computer Science and Engineering,*
*Southern University of Science and Technology, Shenzhen, China*

**Shuxian Li**                                                    *lisx@mail.sustech.edu.cn*
*Department of Computer Science and Engineering,*
*Southern University of Science and Technology, Shenzhen, China*
*Department of Computer Science,*
*Hong Kong Baptist University, Hong Kong SAR, China*

**Xiaoqun Liu**                                                   *Liuxq2021@mail.sustech.edu.cn*
*Department of Computer Science and Engineering,*
*Southern University of Science and Technology, Shenzhen, China*

**Zubin Zheng**                                                   *zhengzb2021@mail.sustech.edu.cn*
*Department of Computer Science and Engineering,*
*Southern University of Science and Technology, Shenzhen, China*

**Siang Yew Chong**                                               *chongsy@sustech.edu.cn*
*Department of Computer Science and Engineering,*
*Southern University of Science and Technology, Shenzhen, China*

**Reviewed on OpenReview:** *https://openreview.net/forum?id=EIPnUofed9*

## Abstract

Existing online class imbalance learning methods fail to achieve optimal performance because their assumptions about enhancing minority classes are hard-coded in model parameters. To learn the model for the performance measure directly instead of using heuristics, we introduce a novel framework based on a dynamic EA called Online Evolutionary Cost Vector (OECV). By bringing the threshold moving method from the cost-sensitive learning paradigm and viewing the cost vector as a hyperparameter, our method transforms the online class imbalance issue into a bi-level optimization problem. The lower layer utilizes a base online classifier for rough prediction, and the upper layer refines the prediction using a threshold moving cost vector learned via a dynamic evolutionary algorithm (EA). OECV benefits from both the efficiency of online learning methods and the high performance of EA, as demonstrated in empirical studies against state-of-the-art methods on thirty datasets. Additionally, we show the effectiveness of the EA component in the ablation study by comparing OECV to its two variants, OECV-n and OECV-ea, respectively. This work reveals the superiority of incorporating EA into online imbalance classification tasks, while its potential extends beyond the scope of the class imbalance setting and warrants future research attention. We release our code[1] for future research.

## 1 Introduction

Online learning from streaming data is common in real-world applications, facing more challenges than offline learning due to limited time and memory resources. Online class imbalance learning involves scenarios

---

[1]https://github.com/t2ance/OECV

where minority classes have notably fewer samples than the majority classes, which can detrimentally affect predictive performance, particularly for minority classes. Current efforts fall into three categories: data-level, algorithm-level, and ensemble approaches. Data-level methods use oversampling and undersampling to rebalance the datasets. Ensemble methods commonly work together with data-level algorithms by randomly resampling incoming data points for each base learner. Algorithm-level approaches react differently to samples from different classes, addressing the tendency to neglect the minority classes.

While designed differently, the three types of methodology all focus on how to efficiently utilize class imbalance information (e.g., imbalance ratio and data distribution) to handle the imbalance issue. However, to our knowledge, they all rely on assumptions about the expected enhancement level for minority classes, which are *ad hoc* and hard-coded in model parameters. For instance, cost-sensitive algorithms, one kind of algorithm-level approach, assign different costs for misclassifying classes based on the class size or performance. But determining optimal costs remains challenging (Liu & Zhou, 2010). In this article, we aim to explore how to achieve optimality concerning any given performance metric directly without making *ad hoc* assumptions. This problem can extend beyond the scope of class imbalance, but we focus on an imbalanced learning setting for simplicity. Gradient-based optimization methods become impractical if we set the non-differentiable evaluation metric as objective. Unfortunately, in fact, a wide range of metrics are non-differentiable since they require a form of step loss function (e.g., counting the number of true positives), which is intractable. This includes accuracy, precision, recall, F1 score, and other comprehensive metrics such as G-mean and balanced accuracy. In an online setting, corresponding prequential evaluation Gama et al. (2014) of the above metrics is used, while the non-differentiability remains. Therefore, we focus on gradient-free optimization methods in this work, particularly the family of EAs (EAs). EAs have been widely studied for classification tasks such as genetic programming (Espejo et al., 2009), learning classifier systems (Sigaud & Wilson, 2007), and evolution of neural networks (Rocha et al., 2007). Besides, recent studies have attempted to leverage EA to assist conventional algorithms in offline class imbalance learning problems (Pei et al., 2023). However, applying EAs to online class imbalance learning remains unexplored and challenging due to time and space constraints in streaming learning. More specifically, evolving classifiers on a large scale and accessing the entire dataset are impossible. Besides, the dynamic environment of concept drift may exist compared to offline learning. To this end, we have to examine a fundamental question: How can we create an online learner that combines two essential traits? That is, it should update fast under a dynamic environment like existing online models while also learning efficiently with non-differentiable objectives similar to EAs.

We propose a novel framework named Online Evolutionary Cost Vector (OECV) to answer this question. OECV is conceptualized as a bi-level optimization problem, with a probabilistic online classifier in the lower layer and a lightweight cost vector in the upper layer. The classifier extracts useful information from data to provide a rough prediction while the cost vector refines the decision boundary. In the case of concept drift, especially the prior drift where class size changes, a dynamic EA is applied to track optimal cost vectors using recent samples contained in a fixed-size buffer. The most crucial difference between our dynamic EA and traditional EA is that it can track the optimal cost vector in a non-stationary environment by maintaining population diversity instead of converging to a certain solution. The computation cost of evolution is drastically reduced since the cost vector to be optimized is lightweight (proportional to the number of classes). In this way, our approach can learn with non-differentiable objectives under dynamic environments and update in a few computation efforts.

The motivation for formulating OECV as a bi-level architecture is highly inspired by the threshold moving method (Kukar et al., 1998; Zhou & Liu, 2005; Sheng & Ling, 2006; Voigt et al., 2014; Hancock et al., 2022) from the paradigm of cost-sensitive learning. The gist of the threshold moving is weighting the probabilistic prediction by the cost vector, which contains the relative cost of misclassifying each class. While the cost vector used in our method acts the same way as that in threshold moving methods, it is usually predefined in the context of cost-sensitive learning. The key point in understanding our motivation is viewing the cost vector as a set of hyperparameters in a class imbalance setting. This would interpret OECV as an online hyperparameter optimization (HPO) method built upon the threshold moving method. The straightforward way of setting the hyperparameter in class imbalance learning is to set it inversely proportional to the class size, which, however, is not guaranteed to be an optimal solution. OECV, on the other hand, manages to optimize the hyperparameters using a dynamic EA on the fly. In viewing OECV as an instance of HPO,

its two levels correspond to searching parameters and hyperparameters separately, where parameters (base classifier) give a rough prediction and hyperparameters (cost vector) refine the prediction. This effectively unifies online class imbalance learning and EAs within a cohesive framework.

The main contributions of this paper are listed as follows:

1. This study is the first to explore the problem of online class imbalance learning using an EA approach. The novel approach OECV unifies EA and online class imbalance learning within a bi-level optimization framework by applying a threshold moving cost vector, effectively addressing the performance-resource trade-off.

2. We present a novel dynamic EA to learn the cost vector under potential concept drift adaptively and incorporate specific prior knowledge about class imbalance to guide evolutionary learning.

3. We study the superiority and efficiency of OECV across thirty real-world datasets. Empirical results show its ability to significantly outperform state-of-the-art (SOTA) methods and confirm the effectiveness of the EA component.

The remainder of this article is organized as follows. Section 2 presents related work. Section 3 details our proposed method. Experimental setup and results are discussed in Section 4. The paper is concluded in Section 5.

## 2 Related Work

Our article is related to online class imbalance learning, threshold moving methods and EA approaches for addressing class imbalance problems.

### 2.1 Online Class Imbalance Learning

Approaches to address online class imbalance problems can typically be classified into three categories, as mentioned in the introduction: data-level, algorithm-level, and ensemble-based methods.

#### 2.1.1 Data-level Methods

Sampling methods work by oversampling and/or undersampling to rebalance data. SMOTE (Chawla et al., 2002) is a synthetic minority over-sampling technique used to balance the class distribution by generating new instances of the minority class. In online learning, it has been adopted in Online SMOTE (Wang & Pineau, 2016), which oversamples using training samples within a sliding window. C-SMOTE (Bernardo et al., 2020) addresses binary class imbalance by actively detecting concept drift via ADWIN (Bifet & Gavalda, 2007), a change detector with a sliding detection window, and applying SMOTE to the minority class in the sliding window. The ignorance of class distribution information results in their sub-optimal performance. OS-CCD based on classification contribution degree is proposed in Jiang et al. (2021), generating synthetic samples via classification contribution degree. SRE (Ren et al., 2019) introduces a selection-based resampling mechanism to handle complex data distributions by considering recent sample properties. However, the resampling procedures of OS-CCD and SRE are both based on clustering, being sensitive to hyperparameters. While showing promising performance, these methods mostly targeted the binary class imbalance problem and needed to maintain a sliding window to reserve relevant training samples, increasing the memory burden.

#### 2.1.2 Algorithm-level Methods

Algorithm-level approaches work by modifying the training process. Qin et al. (2021) employs active learning to select the most important samples to train the classifier. Online one-class Support Vector Machines (Klikowski & Woźniak, 2020) is a kind of one-class classifier that creates a model for each class and achieves a one-class decomposition of multi-class problems. Other algorithm-level methods apply cost-sensitive learning methods, which assign varying costs for misclassifying classes belonging to different classes to reduce the

dominating influence of majority classes, with the common assumption that minority classes incur higher costs. Ksieniewicz (2021) introduces Prior Imbalance Compensation (PIC) for batch learning of imbalanced data streams, which adjusts the decision made by the classifier using class prior probability to compensate for the minority classes. Yan et al. (2017) trains multiple classifiers with various cost matrices and makes predictions by adaptive ensembling. However, it is confined to binary class cases and challenging to extend to multi-class scenarios due to the exponential growth in the number of candidate cost matrices. Other related works (Wang et al., 2021; Ding et al., 2018; Qin et al., 2021) in cost-sensitive methods are based on the weighted extreme learning machine (WELM, Zong et al. (2013)), which is a super efficient single hidden layer neural network with a weighting strategy for class imbalance. WOS-ELM (Wang et al., 2021) integrates a weighting strategy akin to WELM with an online sequential extreme learning machine (OSELM, Huang et al. (2005)). WOS-ELMK (Ding et al., 2018) incorporates kernel mapping, addressing the non-optimal hidden node issue present in WOS-ELM. AI-WSELM (Qin et al., 2021) integrates active learning to significantly reduce labeling costs, demonstrating satisfactory performance compared to existing WELM variants. Despite their promising performance, the weight strategies within this family are explicitly tailored for ELM, limiting their generalizability to other online learning models. In the literature, class sizes are frequently utilized to determine weight strategy. However, this approach does not ensure an optimal weighting schedule.

### 2.1.3 Ensemble Methods

Ensemble methods, such as MOOB, MUOB (Wang et al., 2016), KUE (Cano & Krawczyk, 2020), ROSE (Cano & Krawczyk, 2022), and BEDCOE (Li et al., 2023), effectively tackle the problem by combining resampling techniques. MOOB and MUOB leverage time-decay class size to determine training times. Specifically, the training time for each base classifier is determined by sampling from a Poisson distribution, whose parameter is set according to the class size. The diversity is maintained by random training times on a sample for each base classifier. Kappa Updated Ensemble (KUE) combines online and block-based ensemble approaches and uses Kappa statistics to determine dynamic weighting and select base classifiers. After that, Cano & Krawczyk (2022) proposes an advanced method called ROSE to improve the robustness of KUE by employing adaptive self-tuning, adjusting its parameters, and ensembling the line-up dynamically. To directly deal with class imbalance, ROSE computes the imbalance ratio of each class based on recent samples to derive the training times of each sample. BEDCOE considers potential complex data distribution compared to other works and introduces a borderline enhanced strategy and a disjunct cluster-based oversampling for synthetic sample generation. Despite the improved performance achieved by using multiple base classifiers, the ensemble methods entail a trade-off between the diversity of the ensemble and training time.

### 2.1.4 Common Issues of Existing Methods

We note that the heuristic designs exist in all three categories, and several examples are listed below.

First, some methods suggest the imbalance ratio solely determines the imbalance status and do resampling (Wang & Pineau, 2016; Wang et al., 2016; Bernardo et al., 2020) or design cost schemes (Zong et al., 2013; Wang et al., 2021) based on the estimated online imbalance ratio. However, this is not a unique indicator of class imbalance. Other information, such as data distribution, is also helpful.

Another common assumption is generating synthetic samples around minority instances helps with learning, including Ren et al. (2019), Jiang et al. (2021) and Li et al. (2023). However, this only holds when the minority data is well-clustered and sufficiently discriminative. If the training data is extremely imbalanced or with many corrupted labels, the minority class would be poorly represented and lack a clear structure. In this case, working under this assumption severely jeopardizes the performance.

Additionally, to use the estimated imbalance status such as imbalance ratio or data distribution from clustering, existing works predefine a certain functional form of the relation between imbalance status and training scheme. For instance, WELM (Zong et al., 2013) assumes the cost of misclassifying a class is inversely proportional to its class size. Similarly, MOOB and MUOB (Wang et al., 2016) suppose the training time of one class should be sampled from a Poisson distribution with the imbalance ratio as a parameter. However, the concrete functional form of using imbalance status cannot be exhausted. Besides, none of them have theoretically justified that the proposed functional form could lead to optimality with respect to a certain

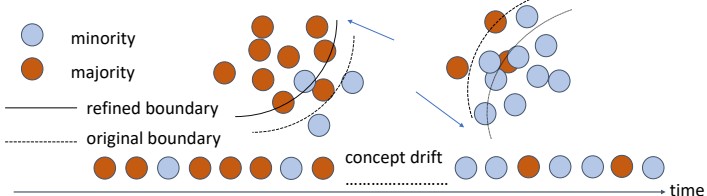

Figure 1: Illustration of the working mechanism of cost vector. The cost vector pushes the decision boundary towards the majority, and the dynamic EA ensures the adaptability of the cost vector for potential concept drift.

performance metric, especially considering that data distribution becomes highly skewed and varies over time. Therefore, we propose to optimize the performance metric directly without fully relying on the estimated imbalance status.

## 2.2 Threshold Moving Method

The threshold moving method (Kukar et al., 1998; Zhou & Liu, 2005; Sheng & Ling, 2006; Voigt et al., 2014; Hancock et al., 2022) is a common technique in cost-sensitive learning. It trains a classifier on the original dataset and prioritizes classes with higher misclassification costs during prediction, using a predefined cost matrix. Formally, denote the cost matrix as $M_{ij}$, where $1 \le i, j \le C$, to represent the cost of misclassifying class $i$ to class $j$. Here $C$ is the number of classes. Let $O_i$, where $1 \le i \le C$, represent the probabilistic output with $\sum_{i=1}^{C} O_i = 1$ and $0 \le O_i \le 1$. The prediction is $\arg\max_i O_i'$ in the threshold moving method comparing to $\arg\max_i O_i$ in standard classifiers, where $O_i'$ is calculated according to

$$O_i' = \eta \sum_{j=1}^{C} O_i M_{ij} = \eta (\sum_{j=1}^{C} M_{ij}) O_i = \eta v_i O_i \tag{1}$$

Here $\eta$ is a normalization term such that $\sum_{i=1}^{C} O_i' = 1$ and $0 \le O_i' \le 1$. A cost vector $v_i = \sum_{j=1}^{C} M_{ij}$ ($1 \le i \le C$) of lower complexity $\mathcal{O}(C)$ can be used in place of the matrix. The cost vector represents the misclassification cost of class $i$ and adjusts the decision boundary toward less costly classes, making it harder to misclassify samples with higher costs. In this paper, the threshold moving is adapted to online class imbalance learning by enabling the cost matrix/vector to be learnable in two novel ways, namely OECV-n and OECV, so that it can respond to the current stream behavior (Fig. 1) rather than being predefined. The baseline OECV-n is designed with time-decay class size, while the main algorithm OECV finds the optimal cost vector based on OECV-n and EA.

## 2.3 EA for Class Imbalance Learning

Recent studies (Pei et al., 2023) have shown the potential of EA in addressing class imbalance, while most of the existing literature remains confined to offline scenarios. In Perry et al. (2015), a genetic algorithm (GA) is used to optimize a class-dependent cost matrix for the weighted updating of a classifier. Sun et al. (2006) introduces a cost-sensitive boosting algorithm that employs GA to optimize a class-dependent cost vector. ECSB (Lemnaru & Potolea, 2017) uses GA to optimize a class-dependent cost matrix and classifier parameters simultaneously. GA is also applied to identify an optimal subset of instances in the majority class (Drown et al., 2009; Khoshgoftaar et al., 2010). In a cost-sensitive SVM method proposed in Cao et al. (2013), the misclassification cost ratio is optimized using particle swarm optimization. Furthermore, differential evolution (DE) has also been tried to optimize class-dependent cost matrices for cost-sensitive deep belief networks (Zhang et al., 2018; 2016). EA is also utilized to support data-level methods. For instance, Jiang et al. (2016) introduces GASMOTE, a GA-based SMOTE approach that optimizes sampling rates for minority class instances.

---

**Algorithm 1:** Training Procedures of Proposed OECV

---

**Input:** Classifier $\mathcal{H}_{t-1}^C$, class size $\Omega_{t-1}$, training sample $(X_t, y_t)$, evolutionary frequency $f$, optimal cost vector $\mathbf{v}^*$, cost vector population $\mathcal{V}$, buffer $\mathcal{B}$

**Output:** Prediction $\hat{y}_t$

**1** Generate rough probabilistic prediction $\mathbf{p}_t$ using $\mathcal{H}_{t-1}^C$.

**2** Produce refined prediction $\mathbf{p}_t^*$ as final prediction $\hat{y}_t$ using $\mathbf{p}_t$ and $\mathbf{v}^*$ by Eqn. 1.

**3** Update $\mathcal{H}_{t-1}^C$ to $\mathcal{H}_t^C$ by its own rule.

**4** Update class size $\Omega_{t-1} \to \Omega_t$ according to Eqn. 3.

**5** Add the sample $(X_t, y_t)$ to $\mathcal{B}$.

**6** **if** $t \mod f == 0$ **then**

**7** $\quad$ Evolve $\mathcal{V}$ by Alg. 2 with $\Omega_t$, $\mathcal{B}$ and $\mathcal{H}_t^C$, and update $\mathcal{V}$ and $\mathbf{v}^*$ based on evolution result.

**8** **return** $\hat{y}_t$

---

There are significant challenges to adapting these methods to online settings. Unlike offline learning, which receives all training data upfront, online learning lacks this comprehensive data overview. Besides, the model must continuously and rapidly adapt to potential concept drift rather than converging. To our knowledge, only Wang & Wang (2023) has adopted a similar idea of EA in online class imbalance learning. It picks base classifiers of different parameter configurations with the highest performance so far. However, characteristics of class imbalance in Wang & Wang (2023) are only used by the original resampling method, and the class imbalance issue is not handled by EA directly. Besides, it is currently tailored for binary classification tasks, making it unsuitable for multi-class scenarios.

## 3  Online Evolutionary Cost Vector (OECV)

In this section, we introduce *Online Evolutionary Cost Vector* (OECV) to illustrate the EA-based cost vector learning approach. Section 3.1 outlines the overall training process. Section 3.2 reformulates the problem into a bi-level optimization. Section 3.3 gives the baseline algorithm OECV-n, and Section 3.4 gives the EA-based algorithm OECV.

### 3.1  Overall Test-then-train Process of OECV

In a data stream $\{(X_t, y_t)\}_{t=1}^{+\infty}$, $X_t \in \mathbb{R}^d$ represents data and $y_t \in \{1, \ldots, C\}$ represents the class label. $C$ is the total number of classes. Uneven class prior distribution leads to class imbalance, and concept drift necessitates the algorithm to adapt to ever-changing data distribution. $X_t$ arrives strictly one by one, being predicted firstly by the latest classifier $\mathcal{H}_{t-1}^C$, and then refined using the cost vector $\mathbf{v}^*$ to give the final prediction $\mathbf{p}_t^*$. $\mathbf{p}_t^*$ is used together with true label $y_t$ that comes before $t + 1$ to update classifier $\mathcal{H}_{t-1}^C$ to $\mathcal{H}_t^C$. This process is known as the test-then-train process.

We present OECV in Alg. 1. At the beginning of the data stream, the cost vector population $\mathcal{V}$ is initialized randomly. At time step $t$, the model $\{\mathcal{H}_{t-1}^C, \mathbf{v}^*\}$, where $\mathcal{H}_t^C$ represents the latest online classifier, and $\mathbf{v}^*$ denotes the optimal cost vector discovered by EA up to time $t - 1$, undergoes initial testing as depicted in Lines 1-2. Here, the online classifier offers an initial prediction, which is then refined by the cost vector. The classifier $\mathcal{H}^C$ updates by its own rule in Line 3. The class size $\Omega_{t-1}$ and fixed-size buffer $\mathcal{B}$ are updated in Lines 4-5, respectively. Cost vector population $\mathcal{V}$ evolves within the **if** statement (Lines 6-7) to yield a new population along with an optimal cost vector $\mathcal{V}^*$. We detail OECV in subsequent subsections individually.

### 3.2  Bi-level Optimization

Due to the impracticality of a full evolution, our framework only evolves partially and breaks down both the model and the problem into two layers (See Fig. 2). The lower layer, being an online classifier, offers a rough probabilistic prediction and updates by its own rule on the fly. The upper layer, being a cost vector, refines the rough prediction and undergoes a dynamic optimization process via dynamic EA. The training data for

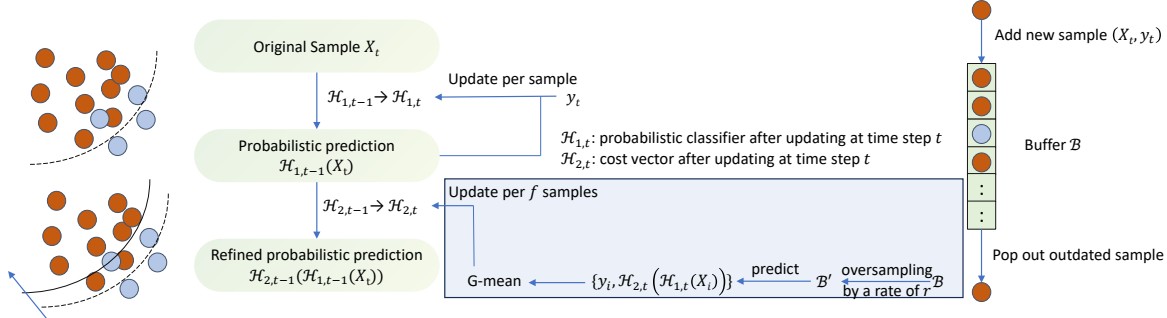

Figure 2: Illustration of OECV as a bi-level optimization problem (Section 3.2). The lower layer consists of a probabilistic classifier, while the upper layer is a cost vector (Section 2.2). A fixed-size buffer is maintained as the training data for the upper level. The cost vector is learned by a dynamic EA at frequency $f$ on the oversampled buffer, using G-mean for objective evaluation (Section 3.4).

updating the cost vector come from a fixed-size buffer $\mathcal{B}$, which is augmented by a simple oversampling trick resulting in a larger buffer $\mathcal{B}'$ to enhance data diversity. As shown in the left part of Fig. 2, we denote the first and upper layers as $\mathcal{H}_{1,t}$ and $\mathcal{H}_{2,t}$, respectively. The complete model is denoted by $\mathcal{H}_t = \{\mathcal{H}_{1,t}, \mathcal{H}_{2,t}\}$. The lower-level problem is to minimize a loss function $\ell_1(\mathcal{H}_1; X_t, y_t)$, which assesses the probabilistic prediction loss computed for each sample in the stream. The upper-level problem involves minimizing a non-differentiable performance metric $\ell_2(\mathcal{H}_{2,t}; p(\cdot; \mathcal{H}_{1,t}^*), y_t)$, which measures the refined prediction error based on the solution $\mathcal{H}_{1,t}^*$ of the lower layer. The learning process of the upper layer occurs at a fixed frequency $f$ instead of updating every time for computational efficiency. Importantly, the lower layer updates solely based on its own rule, and optimizing the upper layer does not affect the lower layer. The overall optimization problem is stated as

$$\min_{\mathcal{H}_{2,t}} \ell_2(\mathcal{H}_{2,t}; p(\cdot; \mathcal{H}_{1,t}^*), y_t) \tag{2}$$
$$s.t. \; \mathcal{H}_{1,t}^* = \arg\min_{\mathcal{H}_{1,t}} \ell_1(\mathcal{H}_1; X_t, y_t)$$

Note that the optimization of the lower level does not depend on the upper level in the sense that $\mathcal{H}_{1,t}^*$ does not depend on $\mathcal{H}_{2,t}$.

In this study, $\mathcal{H}_1$ is parameterized as an online classifier $\mathcal{H}^C$ along with its training loss $\ell_1$ adapted from existing work. $\mathcal{H}^C$ may not consider the specific characteristics of the performance metric (e.g., class imbalance) to be optimized. $\mathcal{H}_2$ is parameterized as a cost vector $\mathbf{v}$, and the choice of $\ell_2$ varies depending on specific needs, such as G-mean or balanced accuracy. In this way, only the upper layer is metric-specific. In the following subsections, we only focus on the learning strategies for the cost vector.

**Remark 1** We notice that the upper level and the lower level are essentially optimized on the overlapping source of data, where the classifier (lower level) uses all data until time $t$, and the cost vector (upper level) uses past sample stores in $\mathcal{B}$. Intuitively, this may result in overfitting in a bi-level optimization problem. In offline bi-level optimization, a better choice is to use distinct training and validation datasets to train the two levels. However, it is more tricky in our online setting since the samples come in the form of a stream. We did not add additional design for simplicity. In fact, the oversampling technique on $\mathcal{B}$, which, although not specifically proposed to handle this problem but proposed to enhance the optimization of the cost vector, may also help. Specifically, this can alleviate the overfitting of overlapping data sources at two levels by introducing diversified data via interpolation, making the data used for the upper level more diversified compared to the lower level.

**Remark 2** Several trade-offs exist in the design of OECV. The first is introducing the updating frequency of cost vector and population size to handle the trade-off between time consumption and performance. Intuitively, a small updating frequency $f$ will allow better performance, and in the extreme case where $f = 1$, the updating frequencies of both levels align, which would achieve the best performance. However, this comes

with high computation costs since updating the cost vector is generally slower than the classifier. Conversely, intermittent updating of the cost vector allows faster training while may incur sub-optimality due to the mismatching of optimization speed of two levels. Similarly, large population size can increase the probability of finding the optimal solution but would induce high updating costs and is not favored in online learning. Another trade-off is between memory consumption and performance, handled by the oversampling rate and buffer size. In practice, large memory allows a large buffer, and an oversampling trick may not be necessary in this case, which is equivalent to setting oversampling rate $r = 1$. In contrast, the oversampling trick can reduce the storage requirement and enhance the data diversity but induces a higher time consumption. Additionally, it may come with issues if label noise exists in $\mathcal{B}$; the labels of augmented buffer $\mathcal{B}'$ are likely to be corrupted as well and result in performance degradation.

Fortunately, the choice of hyperparameters is straightforward and relatively robust within a certain range. In experiments, we set the hyperparameters to make the algorithm as fast as possible while still making improvements compared to baselines. The same set of hyperparameters is used without heavy fine-tuning, which already enables OECV to outperform or be on par with baseline methods over a wide range of datasets. See also Appendix E for an empirical analysis on the sensitivity of hyperparameters above.

### 3.3 Learning Cost Vector with Time Decay Class Sizes

The first approach OECV-n(naive) employs time-decay class sizes (Wang et al., 2018) $\Omega_t = \{\omega_{i,t}\}_{i=1}^{C}$ at time $t$ to continuously track the imbalance status over time using a predefined time decay factor $\lambda$:

$$\omega_{k,t} = \lambda\omega_{k,t-1} + (1 - \lambda) \cdot \mathbb{I}(y_t = k) \qquad\qquad 0 \leq \lambda \leq 1 \qquad\qquad (3)$$

where $\omega_{k,t}$ represents the size of the $k$-th class at time step $t$, and $\mathbb{I}$ is the indicator function. Cost matrix $M_{ij}$ and cost vector $v_i$ are then determined heuristically as follows:

$$M_{ij} = \frac{\omega_{j,t}}{\omega_{i,t}} \qquad\qquad\qquad v_i = \sum_{j=1}^{C} M_{ij} \qquad\qquad (4)$$

In other words, the prediction probability of a class will be scaled up if it is a relative minority class (in the sense of adaptively estimated class size $\omega$) and scaled-down otherwise. The loss function remains untouched in threshold moving, but the prediction probabilities are scaled by $v$. It can adapt to current stream behavior by passively changing the imbalance status. However, optimal performance is not guaranteed as the dependency on the heuristic form of the cost matrix as well as the choice of $\lambda$. The detailed training procedure of OECV-n is similar to that of OECV, by just removing all the use of evolution and replacing $\mathbf{v}^*$ in Alg. 1 by the cost vector determined by Eqn. 4 .

### 3.4 Learning Cost Vector with Dynamic EA

Compared to designing solely with time-decay class size, EAs can find cost vectors that optimize performance measures directly. The evolution process along two related tricks of the resulting OECV are illustrated as follows. The complete algorithm is summarized in Alg. 2.

#### 3.4.1 Evolution

- **Chromosome Encoding**: A cost vector $\mathbf{v}^{(k)}$ is encoded into a chromosome straightforwardly, with the $C$-dimensional vector being the chromosome.

- **Fitness Calculation**: The chance of passing genetic information to subsequent generations relies on the fitness of a cost vector. Recent samples are retained in a fixed-size buffer $\mathcal{B}$ for fitness calculation. $\mathcal{B}$ is enlarged into $\mathcal{B}'$ by oversampling before being used for fitness evaluation. Specifically, we first do classification using the latest classifier $\mathcal{H}^C$ on $\mathcal{B}'$, resulting in the set of rough probabilistic predictions $\{\mathbf{p}_i\}_{i=1}^{|\mathcal{B}'|}$. For each $\mathbf{v}^{(k)}$, it refines the rough predictions to give a set of final predictions $\{\mathbf{p}_i^{(k)}\}_{i=1}^{|\mathcal{B}'|}$. $\{\mathbf{p}_i^{(k)}\}_{i=1}^{|\mathcal{B}'|}$ along with the set of true labels $\{y_i\}_{i=1}^{|\mathcal{B}'|}$ are then used to calculate a performance metric

---

**Algorithm 2:** Cost Vector Evolution

---

**Input:** Buffer $\mathcal{B}$, cost vector population $\mathcal{V}$, online classifier $\mathcal{H}^C$, number of neighbors $k$, sampling rate $r$, size of prior population $m$

**Output:** Optimal cost vector $\mathbf{v}^*$, cost vector population $\mathcal{V}$

**1** // Maintain population diversity and integrate prior knowledge

**2** Generate human-designed cost vector $\mathbf{v}_h$ using $\Omega_t$ by Eqn. 4.

**3** Create prior population $\{\mathbf{v}^{(i)}\}_{i=1}^m$ based on $\mathbf{v}_h$ using Eqn. 5, and add to $\mathcal{V}$.

**4** // Oversampling for data diversity

**5** Initialize augmented buffer $\mathcal{B}'$ with samples from $\mathcal{B}$.

**6 for** $X_t$ *in* $\mathcal{B}$ **do**

**7**     **for** $r-1$ *times* **do**

**8**        Find $k$ nearest neighbors of $X_t$ and randomly select $X_t'$ from them.

**9**        Generate a new sample using Eqn. 6 with $\alpha \sim U(0,1)$, and add it to $\mathcal{B}'$.

**10** // Evolution

**11** Produce rough probabilistic prediction $\{\mathbf{p}_i\}_{i=1}^{|\mathcal{B}'|}$ for each sample in $\mathcal{B}'$ using $\mathcal{H}^C$.

**12** For each $\mathbf{v}^{(k)}$, produce refined predictions $\{\mathbf{p}_i^{(k)}\}_{i=1}^{|\mathcal{B}'|}$ using $\{\mathbf{p}_i\}_{i=1}^{|\mathcal{B}'|}$ (Eqn. 1).

**13** Calculate fitness $f^{(k)}$ for $\mathbf{v}^{(k)}$ based on $\{\mathbf{p}_i^{(k)}\}_{i=1}^{|\mathcal{B}'|}$ and true labels $\{y_i\}_{i=1}^{|\mathcal{B}'|}$.

**14** Evolve $\mathcal{V}$ for one generation by crossover and mutation using $\{f^{(k)}\}_{k=1}^{|\mathcal{V}|}$.

**15** Determine the optimal solution $\mathbf{v}^*$ by comparing fitness.

**16 return** $\mathbf{v}^*$, $\mathcal{V}$

---

as the fitness $f^{(k)}$ of $\mathbf{v}^{(k)}$. With the set of fitness $\{f^{(k)}\}_{k=1}^{|\mathcal{V}|}$, the optimal individual (cost vector) can be determined straightforwardly. Note the performance metric used here is the corresponding offline metric (e.g., G-mean) instead of the online metric (e.g., online G-mean) so that the fitness calculation is not affected by the order of samples in $\mathcal{B}'$.

- **Genetic Operator**: EA employs genetic operators to produce new cost vectors by crossover and mutation based on the fitness value of individuals. Any single objective genetic operator may be applied in the current framework.

If the generation of new cost vectors at Line 14 in Alg. 2 is removed, while the selection of the optimal individual in Line 15 is retained, we get a comparison algorithm OECV-ea as demonstrated in the ablation study. In this case, OECV-ea can be used to show whether OECV works by finding better individuals with evolution instead of simply relying on the buffered data to select a good solution from a large number of candidates.

### 3.4.2 Maintain Population Diversity and Integrate Prior Knowledge

The cost vector designed by time-decay class size as in OECV-n can be used to guide EA. This benefits OECV by integrating the prior knowledge of imbalance status and preventing it from converging to a fixed optimal solution. Specifically, we add $m$ different cost vectors $\{\mathbf{v}^{(i)}\}_{i=1}^m$ randomly generated from $\mathbf{v}_h$ (Eqn. 4):

$$\mathbf{v}^{(i)} = \mathbf{v}_h + \mathbf{w} \qquad\qquad w_j \sim U_j\left(0, \frac{i}{m}\right) \qquad\qquad (5)$$

Recall in the definition of cost vector (Eqn. 1), we require each dimension of $\mathbf{v}^{(i)} = 1$ be in $[0,1]$ and sum up to 1. Therefore, each dimension of $\mathbf{v}^{(i)}$ is clipped to $[0,1]$ and re-normalized. $\{\mathbf{v}^{(i)}\}_{i=1}^m$ are then merged with the previous population to form the initial population for later evolution. After a fixed frequency $f$, the prior population is mixed in, and the population evolves over one generation.

Analogous to the approach with time decay class size in spirits, the dynamic EA also acts passively to counter the effect of concept drift, i.e., not detect the concept drift directly. Although this may not be the best choice

in more complicated scenarios, it would be sufficient for class-prior drift. How to adapt to more complicated drift scenarios is beyond the scope of this work.

### 3.4.3 Oversampling for Data Diversity

To ensure accurate fitness calculation, an oversampling trick for enhancing data diversity is applied to $\mathcal{B}$ to create an augmented buffer $\mathcal{B}'$. Specifically, we expand $\mathcal{B}$ to $r$ times of its original size by generating $r-1$ samples $\{(X_t^{(i)}, y_t^{(i)})\}_{i=1}^{r-1}$ $(r \in \mathbb{N}_+)$ for each $(X_t, y_t) \in \mathcal{B}$:

$$X_t^{(i)} = X_t + \alpha \cdot (X_t' - X_t), \quad y_t^{(i)} = y_t \tag{6}$$

where $\alpha \sim U(0,1)$ and $X_t'$ is randomly selected from $k$ nearest neighbors of $X_t$.

### 3.4.4 Computational Complexity Analysis

We are aware of the potential high computational complexity induced by EAs, including time complexity and space complexity. A formal analysis is provided in this subsection. See also Appendix A for an empirical runtime comparison.

**Memory Complexity Analysis** Denote the population size as $|\mathcal{V}|$ and buffer size as $|\mathcal{B}|$. OECV requires the storage of a buffer of data of size $\mathcal{O}(|\mathcal{B}| \times (d+1))$ where $d$ denotes the number of features, i.e., we count the number of stored samples times the number of features plus one (for the class label). Since temporary synthesized samples in augmented buffer $\mathcal{B}'$ can be processed one by one without storing everything in the memory, whose memory consumption is then negligible, the oversampled data is not taken into account for the extra storage.

**Time Complexity Analysis** The overall time complexity of OECV is linear to the length of the stream, being the same as existing works such as Wang et al. (2016), Qin et al. (2021), and Li et al. (2023). In each time step, the time complexity includes the training cost of both the classifier and the cost vector. The updating of the classifier is a constant and depends on its own rules. For updating of the cost vector, the oversampling on $\mathcal{B}$ takes $\mathcal{O}(|\mathcal{B}| \times (r-1))$ time. Then, $|\mathcal{V}|$ individuals perform prediction and evaluation on $\mathcal{B}'$ that takes $\mathcal{O}(|\mathcal{V}| \times |\mathcal{B}| \times r)$ time in total. The crossover, mutation, and selection operations are based on fitness, being method-dependent. It generally takes $\mathcal{O}(|\mathcal{V}|)$ and is much faster than the fitness calculated in the last step. Summarize and simplify the above steps, and recall the updating of the cost vector occurs at a frequency of $f$, we find the overall time complexity can be represented as $\mathcal{O}(\frac{rT|\mathcal{V}||\mathcal{B}|}{f})$ for the whole data stream, where $T$ is the length of the stream.

While EAs are well-known for their high computational cost (mainly from fitness evaluation), our scheme of applying the cost vector in a post hoc way allows for a much more efficient fitness evaluation. To see this, a forward calculation for prediction is enough to give the fitness, which is attributed to the decoupling of two layers where the training of the classifier is totally independent of the cost vector. Therefore, we only need to evaluate how well the cost vectors correct the current well-trained classifier without any retraining of the classifier. This drastically decreases the time for fitness calculation and makes OECV practical.

### 3.4.5 Storage Requirement

We are aware that the extra storage requirement in the form of a fixed-size buffer is a weakness in our current method. However, certain storage requirements are generally acceptable in the literature, especially widely used in data-level methods (Sec. 2.1.1). The resampling and clustering processes necessitate extra storage similar to ours. For example, Qin et al. (2021), Ren et al. (2019), Cano & Krawczyk (2022), and Wang & Pineau (2016) all require certain storage in the form of a sliding window or chunk. Despite the extra storage burden, it is usually constant and would not increase with the length of the stream. This is practical in many real-world scenarios, such as online edge machine learning, where the stream can be infinitely large, but certain storage (while limited) is accessible. However, if additional storage is unavailable, an adaptive generative model can be used to generate samples to replace the buffer. In this work, we focus on the current extra storage scheme for simplicity.

For a fair comparison in empirical studies, we take AI-WSELM (Qin et al., 2021) (Sec. 4.2) and SmoteOB (Wang & Pineau, 2016) (Appendix F) that require extra storage as compared methods. The extra storage of two methods is set to be larger or comparable than OECV. Additional explorations on whether the performance enhancement is induced by extra data and the influence of the buffer size are also presented in Sec. 4.3 and Appendix E.3.

## 4 Experimental Studies

This section evaluates OECV from several aspects: comparison with SOTA methods, the effectiveness of EA, and the inner working mechanism.

### 4.1 Experimental Setup

We use 30 datasets in total as summarized in Table 1, including 10 streaming datasets (Elec, Abrupt, Gradual, Incremental1, Luxembourg, NOAA, Ozone, Airlines, Covtype, Incremental2, available in the USP-DS repository, Souza et al. (2020)) and 20 real-world offline datasets (remaining 20 datasets in Table 1, available in the Keel repository, Derrac et al. (2015)). The 20 offline datasets are processed in a streaming way to simulate online scenarios. The overall static imbalance ratio for each dataset illustrates the severity of class imbalance, while fluctuation of class imbalance ratio throughout the online learning scenario exists.

Although our method does not require an offline warm start, we use the initial 30% samples of each stream for model initialization in an offline fashion, following the setting in Li et al. (2023). The initialization samples are further split into two datasets in equivalent sizes for training the classifier and the cost vector separately. In this stage, the cost vector population evolves 10 generations to give an initial population for later online training. The buffer size $|\mathcal{B}|$ for OECV is fixed at 200 samples, and the oversampling rate is set to 5 for all datasets. Offline G-mean is used for fitness evaluation on the augmented buffer. The cost vector evolves every 5 sample (i.e., $f = 5$), with the number of individuals set to 25. We employ DE/best/1/L (Opara & Arabas, 2019) as the genetic operator. The implementation of EAs is easy by directly adopting the existing Python packages (such as geatpy [2] used in our experiments). All the hyperparameters related to genetic operators are set to the default values of the existing implementation without tuning. Specifically, the scaling factor of DE is set to 0.5, and exponential crossover is applied with the probability of crossover set to 0.7.

We compare with four SOTA online multi-class imbalance learning methods: MOOB, MUOB (Wang et al., 2016), AI-WSELM (Qin et al., 2021), and BEDCOE (Li et al., 2023). The total number of base learners is set to 10, following (Wang et al., 2016; Li et al., 2023). All methods adhere to the test-then-train process of online learning. Multilayer perceptron serves as the base classifier for all methods, except AI-WSELM, which does not need a base classifier, following the setup in Wang et al. (2016). We set the chunk size (akin to our buffer) of AI-WSELM to be 300, higher than our extra storage of 200. Prequential G-mean with a fading factor of 0.99 is selected as performance metrics, following Wang et al. (2018) and Li et al. (2023). Mean performance across 10 runs is evaluated on the remaining samples after the initialization number. Friedman tests (Demšar, 2006) are used to compare competing methods across datasets statistically. The null hypothesis (H0) posits that all models are equivalent in terms of the predictive performance metric. The alternative hypothesis (H1) suggests that at least one pair of methods differs significantly. If H0 is rejected, the Conover test (Conover & Iman, 1979) is conducted as the post-hoc test.

### 4.2 Performance Comparison

We can see from Table 2(a) that in terms of G-mean, OECV performs the best in 14 out of 30 datasets and the 2nd best in 8 datasets. Friedman tests at significance level 0.05 reject H0 with $p$-value $1.11 \times 10^{-3}$, showing a significant difference between methods. Average ranks (avgRank) across datasets are reported to show how well each method performs compared to others across datasets. The average rank of OECV is 1.967, being the best. Post-hoc tests are then conducted to detect whether OECV has a significant difference

---

[2]https://geatpy.github.io/

Table 1: Overview of the dataset. "#Data" denotes the total number of samples within this dataset, "#Fea" denotes the number of features, "#Class" denotes the number of classes, and IR denotes the overall static imbalance ratio being computed as the ratio between the largest and smallest class sizes.

| Dataset | #Data | #Fea | #Class | IR | Dataset | #Data | #Fea | #Class | IR | Dataset | #Data | #Fea | #Class | IR |
|---------|-------|------|--------|------|----------|-------|------|--------|-------|---------|-------|------|--------|------|
| Elec | 5000 | 8 | 2 | 1.6 | Abalone1 | 2338 | 8 | 2 | 39.3 | Win1 | 691 | 11 | 2 | 68.1 |
| Abrupt | 5000 | 33 | 6 | 4.0 | Abalone2 | 1622 | 8 | 2 | 49.7 | Win2 | 1599 | 11 | 2 | 29.2 |
| Gradual | 5000 | 33 | 6 | 171.2 | Car1 | 1728 | 6 | 2 | 24.0 | Win3 | 656 | 11 | 2 | 35.4 |
| Incremental1 | 5000 | 33 | 6 | 1.0 | Car2 | 1728 | 6 | 2 | 25.6 | Win4 | 1482 | 11 | 2 | 58.3 |
| Luxembourg | 1901 | 31 | 2 | 1.06 | Kddcup | 2225 | 41 | 2 | 100.1 | Win5 | 900 | 11 | 2 | 44.0 |
| NOAA | 5000 | 8 | 2 | 2.4 | Kr | 2901 | 6 | 2 | 26.6 | Yeast1 | 947 | 8 | 2 | 30.6 |
| Ozone | 2534 | 72 | 2 | 14.8 | Segment | 2308 | 19 | 2 | 6.0 | Yeast2 | 1484 | 8 | 10 | 92.6 |
| Airlines | 5000 | 7 | 2 | 2.1 | Shuttle1 | 3316 | 9 | 2 | 66.7 | Yeast3 | 1484 | 8 | 2 | 8.1 |
| Covtype | 5000 | 54 | 7 | 7.0 | Shuttle2 | 1829 | 9 | 2 | 13.9 | Yeast4 | 1484 | 8 | 2 | 32.7 |
| Incremental2 | 5000 | 33 | 6 | 25.4 | Thyroid | 720 | 21 | 3 | 39.2 | Yeast5 | 1484 | 8 | 2 | 41.4 |

from the competitors, for which OECV is chosen as the control method. Post-hoc comparisons show that OECV can significantly outperform all of the competitors.

We can draw two observations on when the OECV can gain an advantage or not from Table 1 and Table 2. Firstly, we notice that when the number of classes is large, e.g., on Gradual, Incremental1, and Yeast2 datasets, our method generally does not perform the best compared to other baselines. Further analysis of the Spearman correlation (Fieller et al., 1957) shows correlation coefficients of 0.49 (moderate) between the number of classes and the value of rank on the 30 datasets (the higher the rank, the worse the performance), being positively correlated. This verifies that our method generally performs better when a small number of classes are presented. This is reasonable since the complexity of the cost vector equals the number of classes, and a larger cost vector is intuitively more difficult to find its optimal solution within limited time and memory. A remedy for this issue deserves a more complicated algorithm design and is left to the future. Secondly, we find our method performs better when the stream is highly skewed, i.e., with a large imbalance ratio. For example, on datasets Win3, Win4, and Win5, our method performs the best among baselines with a larger margin. Similarly, an analysis of the Spearman correlation shows correlation coefficients of $-0.29$ (weak) between the imbalance ratio and the value of rank on the 30 datasets, being negatively correlated, which confirms our conjecture that a highly imbalanced stream favors OECV. We speculate, in this case, the *ad hoc* imbalance estimation, such as the time-decay imbalance ratio (which is used in MOOB, MUOB, and BEDCOE), can not capture the complicated overall imbalance status well. This downgrades the performance of baselines by using a misleading imbalance indicator. In contrast, our method seeks an optimal cost vector directly with respect to the performance metric without consulting heuristically estimated imbalance status. This explains why OECV outperforms other methods under high imbalance.

### 4.3 Ablation Study

Two comparison models OECV-n and OECV-ea have been built in Section 3.3 and Section 3.4, which differ from OECV by just the way on learning cost vector. They are employed here to study the effectiveness of EA. We would expect the performance of OECV, with the full assistance of evolutionary optimization, to be the best. The performance of OECV-ea should be in the middle since while evolution is not used, several candidates of cost vectors are still under consideration for selecting the best one using buffered data. The performance of OECV-n should be the worst because only an imbalance ratio is used. If this occurs, we can conclude that the EA used for optimizing the cost vector is crucial for dealing with class imbalance, and extra data in the buffer is not the determinative reason for performance improvement.

Table 2(b) shows the result in terms of G-mean. The three methods are compared to each other, with Wilcoxon signed rank tests (Wilcoxon, 1992) used to determine if there are significant differences between them. We can see that the average rank of OECV (1.333) is better than that of OECV-n (2.467) and OECV-ea (2.2). Wilcoxon signed rank test rejects H0 with *p*-value 0.0036 and $9.62 \times 10^{-5}$, respectively, meaning OECV is significantly superior to OECV-n and OECV-ea. In comparison between OECV-ea and

Table 2: Performance comparison in terms of G-mean (%). Each entry is the mean±std performance across 10 runs. The best performance on each dataset is highlighted in **bold**, and the 2nd best performance is highlighted in *italics*. The last row lists the average ranks (avgRank) of each model across datasets in each subtable. Part (a) compares SOTA methods and the proposed OECV. A significant difference against OECV is highlighted in yellow. Part (b) reports the ablation results between variants of OECV.

(a) Performance comparison

| Dataset | AI-WSELM | MOOB | MUOB | BEDCOE | OECV |
|---|---|---|---|---|---|
| Elec | 78.2±1.6 | *90.9±0.2* | 88.7±0.4 | **95.5±0.1** | 83.7±0.9 |
| Abrupt | **66.2±1.4** | 60.2±1.7 | 60.4±2.2 | 60.0±0.4 | *62.8±0.6* |
| Gradual | 0.0±0.0 | *22.4±9.1* | 0.1±0.3 | **34.8±20.4** | 8.5±4.2 |
| Incremental1 | 46.0±0.7 | **53.8±0.6** | 48.5±2.0 | *52.9±0.4* | 46.4±1.5 |
| Luxembourg | 85.5±2.4 | **100.0±0.0** | **100.0±0.0** | **100.0±0.0** | **100.0±0.0** |
| NOAA | *71.3±0.8* | 65.3±0.7 | 64.6±0.6 | 68.2±0.7 | **73.1±0.5** |
| Ozone | 65.0±2.9 | 72.3±1.8 | **78.0±0.6** | 70.6±1.3 | *77.1±1.7* |
| Airlines | *50.8±1.0* | 34.6±2.8 | 47.6±1.6 | 50.6±0.4 | **51.8±0.9** |
| Covtype | 0.0±0.0 | **65.4±0.8** | 0.0±0.0 | *64.6±1.2* | 28.6±1.5 |
| Incremental2 | 0.8±0.2 | *30.8±5.0* | 0.9±1.2 | **40.9±1.6** | 15.6±1.6 |
| Abalone1 | 43.6±5.2 | 55.0±0.9 | *64.5±3.8* | 59.1±0.8 | **67.8±4.3** |
| Abalone2 | **48.0±9.3** | 4.6±0.0 | 26.8±8.2 | 33.2±0.0 | *38.7±7.6* |
| Car1 | **80.4±2.9** | 33.3±4.3 | 56.3±4.9 | 44.5±5.0 | *78.2±2.2* |
| Car2 | **96.5±2.9** | 74.9±0.7 | 79.8±3.7 | 74.4±1.5 | *96.1±1.0* |
| Kddcup | 78.1±11.8 | **100.0±0.0** | 95.9±3.5 | **100.0±0.0** | **100.0±0.0** |
| Kr | 94.3±1.7 | *94.4±0.7* | 90.5±1.8 | 90.2±0.7 | **94.7±1.3** |
| Segment | 98.7±0.4 | 98.9±0.1 | 93.0±0.6 | *99.0±0.0* | **99.1±0.1** |
| Shuttle1 | **100.0±0.0** | 99.4±0.6 | 97.9±1.7 | 99.0±0.9 | *99.9±0.0* |
| Shuttle2 | 99.4±0.1 | 99.6±0.0 | **99.8±0.1** | *99.7±0.1* | *99.7±0.0* |
| Thyroid | 29.0±14.0 | 38.9±2.7 | 0.6±0.0 | *56.7±2.2* | **71.6±1.5** |
| Win1 | 29.1±34.1 | 6.8±0.0 | 6.8±0.0 | *36.5±36.4* | **80.6±1.2** |
| Win2 | 39.0±4.8 | 15.4±5.1 | **62.0±5.0** | 27.3±1.4 | *59.2±3.6* |
| Win3 | 26.2±11.3 | 22.1±2.4 | 19.6±10.0 | *26.5±2.8* | **79.9±1.2** |
| Win4 | 9.7±10.7 | *43.2±11.6* | 16.7±9.5 | 27.7±1.6 | **50.6±5.7** |
| Win5 | 22.8±19.3 | *32.8±4.7* | 11.0±4.2 | 14.7±0.0 | **53.3±7.1** |
| Yeast1 | *47.8±8.0* | 32.0±0.5 | 36.1±13.8 | 33.2±4.1 | **48.0±18.9** |
| Yeast2 | **28.0±6.0** | 0.1±0.1 | 0.0±0.0 | *8.8±4.2* | 0.2±0.4 |
| Yeast3 | 81.5±2.3 | *89.2±0.3* | **89.8±1.2** | 87.5±0.3 | 87.8±1.0 |
| Yeast4 | 72.9±6.6 | **86.5±0.8** | *82.4±9.5* | 71.7±2.8 | 81.7±5.7 |
| Yeast5 | *70.3±3.8* | 64.1±1.9 | 51.7±7.1 | 53.1±3.0 | **86.5±1.7** |
| avgRank | 3.35 | 3.133 | 3.517 | 3.033 | 1.967 |

(b) Ablation studies

| OECV-n | OECV | OECV-ea |
|---|---|---|
| 83.1±0.4 | **83.7±0.9** | *83.6±0.9* |
| 62.0±0.7 | **62.8±0.6** | *62.6±0.9* |
| **15.8±2.5** | *8.5±4.2* | 4.3±2.8 |
| 45.9±1.2 | **46.4±1.5** | *46.2±1.2* |
| **100.0±0.0** | **100.0±0.0** | **100.0±0.0** |
| *73.0±0.5* | **73.1±0.5** | 72.9±0.5 |
| 71.8±1.9 | **77.1±1.7** | *76.1±1.6* |
| 50.7±0.5 | **51.8±0.9** | *51.7±0.8* |
| **38.7±1.1** | *28.6±1.5* | 26.6±1.3 |
| **21.2±1.8** | *15.6±1.6* | 13.0±1.0 |
| 55.7±3.0 | **67.8±4.3** | *59.0±5.7* |
| 25.6±0.1 | **38.7±7.6** | *33.1±5.9* |
| 77.0±3.1 | **78.2±2.2** | *77.5±2.0* |
| 94.7±1.2 | **96.1±1.0** | *95.2±0.9* |
| **100.0±0.0** | **100.0±0.0** | **100.0±0.0** |
| 91.5±0.8 | **94.7±1.3** | *92.2±1.3* |
| *99.1±0.1* | *99.1±0.1* | **99.4±0.1** |
| **99.9±0.0** | **99.9±0.0** | **99.9±0.0** |
| **99.7±0.0** | **99.7±0.0** | 99.6±0.0 |
| 68.5±3.8 | *71.6±1.5* | **73.6±2.2** |
| 79.9±0.1 | **80.6±1.2** | *80.6±0.9* |
| 48.1±0.9 | **59.2±3.6** | *49.6±1.9* |
| 39.0±4.2 | **79.9±1.2** | *60.6±12.9* |
| 18.9±7.7 | **50.6±5.7** | *29.1±4.1* |
| *53.3±5.6* | **53.3±7.1** | 41.0±7.4 |
| *16.5±0.9* | **48.0±18.9** | 15.5±0.2 |
| *0.0±0.0* | **0.2±0.4** | *0.0±0.0* |
| 85.0±1.1 | **87.8±1.0** | *86.8±0.8* |
| 68.3±4.0 | **81.7±5.7** | *75.0±3.7* |
| *84.8±0.1* | **86.5±1.7** | 81.8±2.8 |
| 2.467 | 1.333 | 2.2 |

OECV-n, the average rank of OECV-ea (2.2) is better than OECV-n (2.467), but the Wilcoxon signed rank test fails to reject H0 with $p$-value 0.178, meaning there is no significant difference between OECV-ea and OECV-n. From the comparison between OECV and OECV-n, we see that eliminating the whole EA strategy would significantly decline performance. Additionally, we can see from the comparison between OECV-ea and OECV, as well as OECV-ea and OECV-n, that the extra data does not play a determinative role. This is because OECV-ea also uses extra data in finding optimal cost vector with the performance set to the objective of performance metric, but it does not perform significantly better than OECV-n and performs significantly worse than OECV. This implies that it is the EA instead of extra data making OECV outperform compared methods.

### 4.4 Further Discussions

We explore two related questions to assess the working mechanism of OECV.

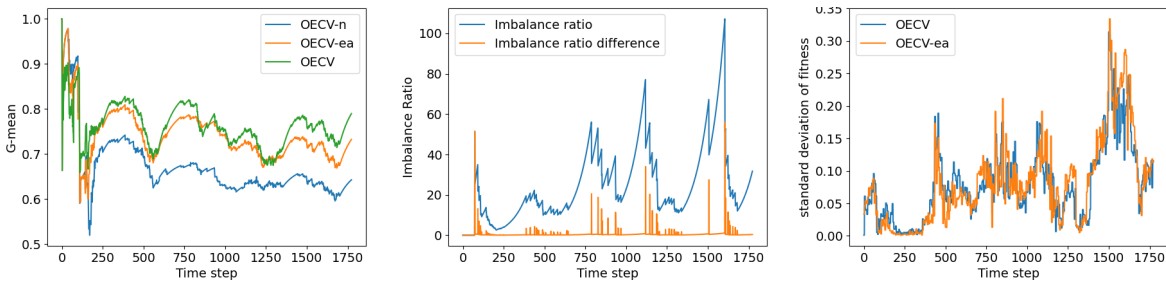

Figure 3: Prequential G-mean, imbalance ratio, and standard deviation of fitness of the population in dataset Ozone. The higher the standard deviation, the greater the diversity. Imbalance ratio is calculated by time-decay class sizes (Eqn. 3).

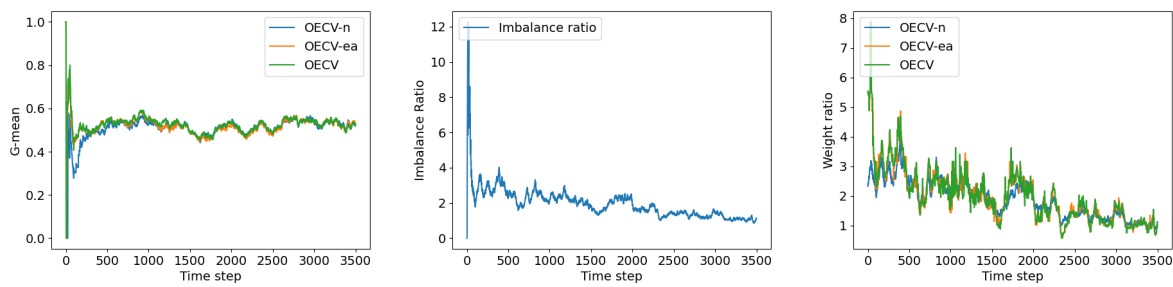

Figure 4: Prequential G-mean, imbalance ratio, and weight ratio in dataset Airlines. Imbalance ratio is calculated by time-decay class sizes (Eqn. 3).

### 4.4.1 Analysis on Population Diversity

We explore whether OECV can maintain population diversity over time instead of converging. The population diversity enables OECV to track the optimal cost vector instead of converging to a certain solution. We present the standard derivation of individual fitness in Fig. 3 with a further analysis of the Spearman correlation (Fieller et al., 1957). The result shows correlation coefficients of 0.594 (moderate) and 0.629 (strong) between the absolute difference of imbalance ratios (i.e., the absolute value of the difference between two neighboring class imbalance ratios) and standard deviation (std) of fitness of OECV and OECV-ea, respectively, being positively correlated. It also shows a high correlation coefficient of 0.869 between the std of fitness of OECV and OECV-ea. We can draw two conclusions: 1) The diversity adapts to data stream behavior. This means OECV and OECV-ea can expand the exploration of new cost vectors (high std) during a concept drift where the imbalance ratio changes drastically while adopting temporary elitists by leveraging learned knowledge about class imbalance (low std) during the steady stream where the imbalance ratio is stable. 2) Despite similar diversity and changing behavior, OECV outperforms OECV-ea. This indicates the superiority of EA in that it can maintain a population of cost vectors with higher quality under the same diversity.

### 4.4.2 Analysis on EA-based Cost Vector

We explore how the cost vector found by the EA outperforms the one determined solely by the imbalance ratio. We define the weight ratio (WR) as $\frac{v_1}{v_0}$ to visualize the cost vector in a binary classification scenario in Fig. 4. Here, $v_i$ represents the $i$-th dimension of the cost vector. Analogous to the imbalance ratio, the WR serves as a belief of the imbalance level indicated by the cost vector. We analyze the Spearman correlation between the WR of three variants and the imbalance ratio, yielding correlation coefficients of 0.971, 0.897, and 0.887 for OECV-n, OECV-ea, and OECV respectively, indicating strong correlations. This means the cost vectors found by EA can also reflect the beliefs about class imbalance, while some of these beliefs are

sacrificed to seek more appropriate values of the cost vector in finding the optimal solution. Besides, Fig. 4 illustrates the challenge of finding the optimal solution by *ad hoc* assumptions: While OECV outperforms that of OECV-n, the WR of OECV fluctuates compared to OECV-n. This suggests that relying solely on the imbalance ratio is insufficient for identifying the best cost vector. The dynamic EA addresses this limitation by directly setting the performance measure as the objective and avoiding heuristic reliance.

## 5    Conclusion

This article introduces a novel approach Online Evolutionary Cost Vector (OECV) to tackle the online class imbalance issue by eliminating heuristic assumptions on class imbalances widely used in existing methods. OECV instead manages to optimize performance on any specified performance metrics directly, achieved by adopting a dynamic EA. The model is explicitly deconstructed into two layers: an online classifier for rough probabilistic prediction and a cost vector for refining the decision boundary. The cost vector is the only part subject to the dynamic EA for directly optimizing specific performance metrics. This bi-level architecture is motivated by viewing the cost vector as a hyperparameter in the threshold moving method and the EA as an approach to fine-tune the hyperparameter. A dynamic EA is employed to track the optimal cost vector over time. Cost vectors designed by class size are integrated into the prior population to sustain population diversity and integrate prior knowledge. To enhance data diversity, an oversampling trick is used to augment the buffer and attain more beneficial evolutionary results. Empirical studies demonstrate the validity and efficiency of our approach. Analysis of the working mechanism reveals how OECV can generate a superior cost vector compared to the human-designed counterpart.

The potential of the OECV framework extends beyond the class imbalance setting and has further exploration values in various other classification tasks. High performance across a broad range of metrics unrelated to class imbalance could be achieved with only slight adjustments to the cost vector. For instance, OECV can simultaneously serve multi-objective purposes by optimizing for multiple metrics, including accuracy, recall, and F1-score. Another future work is to handle the potential label noise. Specifically, when there exist corrupted labels, the samples in the augmented buffer will also contain corrupted labels, which may degrade the optimization of cost vectors and deserve a further specific design.

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

## A  Running Time Comparison

Since EA is known for high time complexity, we conduct a runtime comparison to show the practicality of OECV along with the theoretical analysis in Sec. 3.4.4. All experiments are benchmarked on a server configured with Intel(R) Xeon(R) Gold 6338 CPU @ 2.00GHz. The geometric mean of runtime across datasets is reported in the case of varying runtime scales across datasets. Specifically, suppose $N$ datasets are used, we report $\sqrt[N]{\prod_{i=1}^{N} t_i}$, where $t_i$ represents the runtime on the $i$-th dataset. Two key observations can be made from the results in Table 3. Firstly, while some methods exhibit significantly shorter runtimes, such as MUOB and OECV-n, this comes at the expense of their inferior performance, as evidenced in Table 2(a). Secondly, our approach demonstrates remarkable efficiency, as OECV achieves the best rank with tolerable runtime compared to other SOTA methods. This validates the time efficiency and practicality of OECV despite integrating EA.

## B  Performance Comparison in Terms of Balanced Accuracy

In Table 4 (a), we include a performance comparison in terms of balanced accuracy, complementary to the results in Section 4.2. We can see from Table 4 that in terms of balanced accuracy, OECV performs the best in 12 out of 30 datasets and the 2nd best in 8 data sets. Friedman tests (Demšar, 2006) at the significance level 0.05 reject H0 with the $p$-value $4.21 \times 10^{-3}$, showing that there is a significant difference between methods. The average rank of OECV is 2.167, being the best. Post-hoc tests are then conducted to investigate whether OECV has a significant difference from the competitors, for which OECV is chosen as the control method. Post-hoc comparisons show that OECV can significantly outperform all of the competitors except BEDCOE, where the $p$-value is 0.052, being only marginally higher than 0.05. We conjecture this is because the optimization objective is set to G-mean instead of balanced accuracy in OECV, making the algorithm not aware of this performance metric.

## C  Ablation Studies in Terms of Balanced Accuracy

Table 4 (b) shows the predictive performance of the three models in terms of balanced accuracy, complementary to the results in Section 4.3. Then, the three methods are compared to each other, with Wilcoxon signed rank tests (Wilcoxon, 1992) used to determine if there are significant differences between them.

We can see that in terms of balanced accuracy, the average rank of OECV (1.55) is better than that of OECV-n (2.233) and OECV-ea (2.217). Wilcoxon signed rank test rejects H0 with $p$-value 0.042 and $4.98 \times 10^{-4}$, respectively, meaning OECV is significantly superior to OECV-n and OECV-ea. This indicates that eliminating the EA strategy would significantly decline predictive performance in terms of balanced accuracy, showing its effectiveness.

We follow a similar procedure to compare OECV-ea and OECV-n. In terms of balanced accuracy, the average rank of OECV-ea (2.217) is better than OECV-n (2.233). Wilcoxon signed-rank test fails to reject H0 with $p$-value 0.838, meaning there is no significant difference between OECV-ea and OECV-n. This indicates that using extra samples in the buffer is solely insufficient to find a significantly better cost vector. In other words,

Table 3: Comparison between methods in terms of runtime in seconds. The geometric mean of runtime is shown in the last row.

| Dataset | AI-WSELM | MOOB | MUOB | BEDCOE | OECV-n | OECV | OECV-ea |
|---|---|---|---|---|---|---|---|
| Elec | 6.9±1.2 | 49.8±1.4 | 22.5±0.6 | 134.2±3.4 | 6.5±0.1 | 47.5±0.3 | 53.6±3.9 |
| Abrupt | 13.3±2.6 | 51.2±1.5 | 11.2±0.2 | 292.8±20.3 | 6.5±0.3 | 81.4±3.4 | 75.5±1.2 |
| Gradual | 15.0±3.2 | 52.9±0.4 | 5.1±0.1 | 212.8±5.1 | 6.7±0.2 | 110.3±20.5 | 74.9±1.4 |
| Incremental1 | 15.1±3.2 | 49.7±0.2 | 16.2±0.4 | 383.4±5.8 | 6.5±0.2 | 101.5±2.9 | 76.1±1.2 |
| Luxembourg | 5.3±0.1 | 16.4±0.1 | 12.6±0.1 | 28.2±0.5 | 2.5±0.3 | 38.0±2.1 | 27.6±0.7 |
| NOAA | 10.0±0.2 | 48.4±0.2 | 23.8±0.2 | 307.8±13.2 | 7.0±0.7 | 53.9±0.6 | 49.6±0.7 |
| Ozone | 8.0±0.1 | 38.5±1.7 | 6.3±0.3 | 139.4±56.7 | 4.5±0.1 | 55.9±1.8 | 42.7±0.7 |
| Airlines | 12.9±0.4 | 49.0±0.9 | 25.8±0.3 | 364.2±6.4 | 7.3±0.5 | 52.8±0.6 | 47.1±0.4 |
| Covtype | 27.8±0.3 | 58.7±1.0 | 5.3±0.0 | 252.7±4.7 | 7.3±0.2 | 115.4±1.7 | 80.6±1.0 |
| Incremental2 | 17.5±0.5 | 62.4±0.7 | 5.6±0.0 | 379.0±7.1 | 7.2±0.2 | 107.5±3.2 | 75.1±0.9 |
| Abalone1 | 2.4±0.0 | 24.2±0.4 | 4.0±0.2 | 73.4±3.1 | 3.3±0.1 | 25.1±0.4 | 22.7±0.3 |
| Abalone2 | 1.6±0.0 | 16.0±0.4 | 2.5±0.1 | 52.2±1.6 | 2.4±0.2 | 17.4±0.6 | 15.6±0.2 |
| Car1 | 1.6±0.0 | 22.5±0.8 | 4.6±0.8 | 76.5±5.0 | 2.5±0.1 | 18.3±0.3 | 16.4±0.2 |
| Car2 | 1.6±0.0 | 21.3±0.4 | 3.7±0.6 | 46.2±0.7 | 2.5±0.1 | 18.6±0.5 | 16.3±0.1 |
| Kddcup | 7.3±0.3 | 21.1±0.2 | 3.1±0.1 | 36.4±0.8 | 3.3±0.1 | 43.9±2.4 | 31.9±0.6 |
| Kr | 3.8±0.1 | 36.4±0.8 | 5.7±0.4 | 65.9±0.8 | 4.3±0.2 | 30.8±0.6 | 27.4±0.3 |
| Segment | 6.3±0.2 | 31.1±0.6 | 8.9±0.3 | 52.6±1.8 | 3.5±0.4 | 41.4±1.8 | 29.4±0.8 |
| Shuttle1 | 6.4±0.4 | 39.0±1.2 | 6.0±0.5 | 55.4±0.9 | 4.9±0.2 | 38.1±0.5 | 32.5±0.4 |
| Shuttle2 | 3.8±0.2 | 22.9±0.6 | 4.5±0.1 | 31.9±0.8 | 2.8±0.6 | 21.2±0.5 | 17.7±0.2 |
| Thyroid | 2.0±0.1 | 10.1±0.1 | 1.0±0.1 | 37.1±0.6 | 1.2±0.3 | 12.2±0.8 | 8.5±0.3 |
| Win1 | 1.5±0.1 | 7.7±0.2 | 0.9±0.1 | 19.4±0.3 | 1.0±0.2 | 8.4±0.3 | 6.8±0.1 |
| Win2 | 3.6±0.2 | 17.0±0.9 | 3.0±0.2 | 64.6±1.2 | 2.3±0.2 | 18.6±0.5 | 15.5±0.2 |
| Win3 | 1.4±0.1 | 8.1±0.4 | 1.2±0.4 | 23.2±1.1 | 1.0±0.1 | 7.5±0.3 | 6.4±0.1 |
| Win4 | 3.3±0.1 | 16.7±0.4 | 2.2±0.4 | 37.2±1.0 | 2.1±0.1 | 17.0±0.3 | 14.3±0.1 |
| Win5 | 2.0±0.1 | 11.1±0.6 | 1.3±0.1 | 27.7±0.9 | 1.3±0.1 | 10.4±0.3 | 8.7±0.1 |
| Yeast1 | 1.9±0.1 | 13.1±0.8 | 1.7±0.3 | 36.5±0.8 | 1.4±0.1 | 10.2±0.3 | 9.2±0.1 |
| Yeast2 | 3.2±0.1 | 22.5±0.5 | 1.8±0.3 | 150.2±5.2 | 2.1±0.3 | 20.7±0.4 | 15.8±0.1 |
| Yeast3 | 3.1±0.1 | 17.6±0.3 | 4.7±0.2 | 57.5±1.0 | 2.1±0.1 | 15.7±0.2 | 14.5±0.1 |
| Yeast4 | 3.1±0.2 | 17.7±0.5 | 2.4±0.2 | 40.4±0.6 | 2.3±0.6 | 15.6±0.2 | 14.3±0.2 |
| Yeast5 | 3.0±0.1 | 17.4±0.4 | 2.3±0.3 | 42.5±0.7 | 2.1±0.2 | 15.9±0.5 | 14.2±0.1 |
| G-mean time | 4.501 | 24.471 | 4.378 | 75.8 | 3.074 | 28.517 | 23.66 |

although our method uses extra data, this is not the determinative reason why OECV can outperform SOTA methods.

## D Continuous Performance Throughout Time

Figure 5 presents performance comparisons over various time steps on two representative datasets in terms of G-mean and balanced accuracy. Similar patterns were observed in other datasets. We can see that OECV consistently outperforms most other methods across most time steps in terms of both G-mean and balanced accuracy. This demonstrates the continuous effectiveness of our approach in improving performance over time.

For ablation studies, we demonstrate continuous performance over time in Figure 6 in terms of G-mean and balanced accuracy. We notice removing the evolutionary cost vector strategy leads to a continual decline in performance across most test steps. As a result, we assert that using EA is crucial in our approach.

Table 4: Performance comparison in terms of balanced accuracy (%). Each entry is the mean±std performance across 10 runs. The best performance on each dataset is highlighted in **bold**, and the 2nd best performance is highlighted in *italics*. The last row lists the average ranks (avgRank) of each model across datasets in each subtable. Part (a) reports the comparison between SOTA methods and the proposed OECV. A significant difference against OECV is highlighted in yellow. Part (b) reports the ablation results between variants of OECV.

(a) Performance comparison

| Dataset | AI-WSELM | MOOB | MUOB | BEDCOE | OECV |
|---|---|---|---|---|---|
| Elec | 79.5±1.7 | *91.0±0.2* | 88.9±0.4 | **95.5±0.1** | 84.2±0.9 |
| Abrupt | **68.2±1.1+** | 65.9±0.6 | *67.3±0.6* | 63.7±0.3 | 64.6±0.6 |
| Gradual | 34.1±0.4 | *63.1±0.4* | 20.2±0.8 | **64.1±1.3** | 52.3±1.0 |
| Incremental1 | 48.5±0.6 | **58.6±0.4** | *58.1±0.6* | 54.7±0.4 | 48.2±1.3 |
| Luxembourg | 85.6±2.4 | **100.0±0.0** | **100.0±0.0** | **100.0±0.0** | **100.0±0.0** |
| NOAA | *72.0±0.7* | 66.0±0.6 | 64.9±0.6 | 69.0±0.6 | **73.2±0.5** |
| Ozone | 67.7±2.2 | 74.9±1.3 | **78.3±0.6** | 74.1±1.0 | *77.4±1.6* |
| Airlines | 51.6±0.6 | 51.6±0.5 | 51.1±0.7 | *51.7±0.4* | **52.2±0.8** |
| Covtype | 21.1±2.9 | **70.6±0.4** | 16.4±3.0 | *70.3±0.9* | 38.6±1.1 |
| Incremental2 | 30.1±0.5 | *49.3±0.4* | 25.2±1.9 | **49.4±1.0** | 40.0±0.6 |
| Abalone1 | 60.6±2.3 | 65.6±0.6 | 66.5±2.9 | 68.0±0.5 | **71.9±2.5** |
| Abalone2 | **61.0±3.3** | 51.6±0.2 | 45.2±3.4 | *56.2±0.2* | 54.2±3.1 |
| Car1 | **82.1±2.5** | 53.7±1.1 | 62.4±6.0 | 57.4±2.2 | *79.0±2.1* |
| Car2 | **96.7±2.6** | 77.3±0.5 | 81.2±3.7 | 77.7±1.1 | *96.2±1.0* |
| Kddcup | 83.5±7.3 | **100.0±0.0** | 96.1±3.3 | **100.0±0.0** | **100.0±0.0** |
| Kr | *94.5±1.6* | 94.4±0.7 | 91.0±1.6 | 90.6±0.7 | **94.7±1.2** |
| Segment | 98.7±0.4 | 98.9±0.1 | 93.3±0.6 | *99.0±0.0* | **99.1±0.1** |
| Shuttle1 | **100.0±0.0** | 99.4±0.6 | 98.0±1.7 | 99.1±0.9 | *99.9±0.0* |
| Shuttle2 | 99.4±0.1 | 99.6±0.0 | **99.8±0.1** | *99.7±0.1* | 99.7±0.0 |
| Thyroid | 54.6±4.6 | 53.9±2.8 | 34.7±0.0 | *63.9±1.9* | **75.0±1.5** |
| Win1 | 62.6±14.6 | 53.0±0.1 | 53.4±0.0 | *65.5±15.7* | **83.3±0.5** |
| Win2 | 54.9±1.7 | 51.0±0.9 | **65.0±2.5** | 52.6±0.6 | *64.6±2.0* |
| Win3 | *52.4±2.8* | 51.8±0.8 | 51.5±3.5 | 52.1±1.0 | **80.2±1.2** |
| Win4 | 52.7±1.8 | **59.8±4.6** | 50.1±2.3 | 54.4±0.5 | *59.5±4.4* |
| Win5 | *55.5±5.0* | 53.3±2.0 | 49.4±1.3 | 49.6±0.3 | **58.5±3.8** |
| Yeast1 | **60.1±3.7** | 56.7±0.4 | 53.1±5.8 | 57.1±1.0 | *59.0±7.7* |
| Yeast2 | **45.7±2.7** | 39.4±2.6 | 10.8±1.7 | *41.2±0.8* | 39.9±1.1 |
| Yeast3 | 82.3±2.0 | *89.3±0.3* | **90.1±1.0** | 87.8±0.3 | 87.9±1.0 |
| Yeast4 | 76.7±4.8 | **88.9±0.8** | *84.6±7.4* | 76.7±2.0 | 83.4±4.3 |
| Yeast5 | *75.3±2.4* | 73.5±1.1 | 65.8±4.3 | 66.1±1.9 | **86.8±1.6** |
| avgRank | 3.1 | 3.1 | 3.717 | 2.917 | 2.167 |

(b) Ablation studies

| OECV-n | OECV | OECV-ea |
|---|---|---|
| 83.7±0.4 | **84.2±0.9** | *84.1±0.8* |
| 64.4±0.7 | *64.6±0.6* | **64.7±0.8** |
| **59.2±0.9** | *52.3±1.0* | 50.6±1.1 |
| 47.9±1.1 | **48.2±1.3** | *48.0±1.1* |
| **100.0±0.0** | **100.0±0.0** | **100.0±0.0** |
| *73.1±0.5* | **73.2±0.5** | 73.0±0.5 |
| 73.6±1.4 | **77.4±1.6** | *76.7±1.5* |
| 51.6±0.6 | **52.2±0.8** | **52.2±0.9** |
| **50.1±1.1** | *38.6±1.1* | 33.9±1.2 |
| **43.2±0.6** | *40.0±0.6* | 36.4±0.7 |
| 65.8±1.4 | **71.9±2.5** | *67.2±2.7* |
| 54.0±0.3 | *54.2±3.1* | **54.4±1.8** |
| **79.1±2.4** | *79.0±2.1* | 78.8±1.7 |
| 94.9±1.1 | **96.2±1.0** | *95.3±0.9* |
| **100.0±0.0** | **100.0±0.0** | **100.0±0.0** |
| 91.8±0.7 | **94.7±1.2** | *92.5±1.2* |
| *99.1±0.1* | 99.1±0.1 | **99.4±0.1** |
| **99.9±0.0** | **99.9±0.0** | **99.9±0.0** |
| **99.7±0.0** | 99.7±0.0 | 99.6±0.0 |
| 73.1±2.7 | *75.0±1.5* | **77.1±2.1** |
| *83.5±0.1* | 83.3±0.5 | **83.8±0.6** |
| *60.3±0.4* | **64.6±2.0** | 59.7±1.2 |
| 56.4±2.2 | **80.2±1.2** | *66.0±7.7* |
| *51.2±1.5* | **59.5±4.4** | 51.1±2.2 |
| **63.8±2.7** | *58.5±3.8* | 54.8±4.3 |
| 53.2±0.3 | **59.0±7.7** | 51.6±0.7 |
| **40.0±1.7** | *39.9±1.1* | 36.4±1.7 |
| 85.5±1.0 | **87.9±1.0** | *87.0±0.7* |
| 73.8±3.0 | **83.4±4.3** | *78.5±2.7* |
| *85.7±0.1* | **86.8±1.6** | 83.2±2.2 |
| 2.233 | 1.55 | 2.217 |

# E  Hyperparameter Analysis

To balance the performance and computational cost, we introduced a few hyperparameters in OECV. The role of each hyperparameter is straightforward and does not need heavy fine-tuning. In this section, we provide a detailed discussion on the sensitivity of population size, oversampling rate, buffer size, and the updating frequency of the cost vector. We also investigate the influence of the pre-training ratio. Note the pre-training stage is not necessary in our method and is added to make a fair comparison since Li et al. (2023) requires a pre-training setup. The ratio of 30% in the main experiments is chosen randomly and set to be the same for all compared methods without any tuning.

## E.1  Population Size

We include a further experiment on the sensitivity of population size setting in OECV. Fixing the other original hyperparameter settings of OECV, we manually alter only the population size (i.e., number of

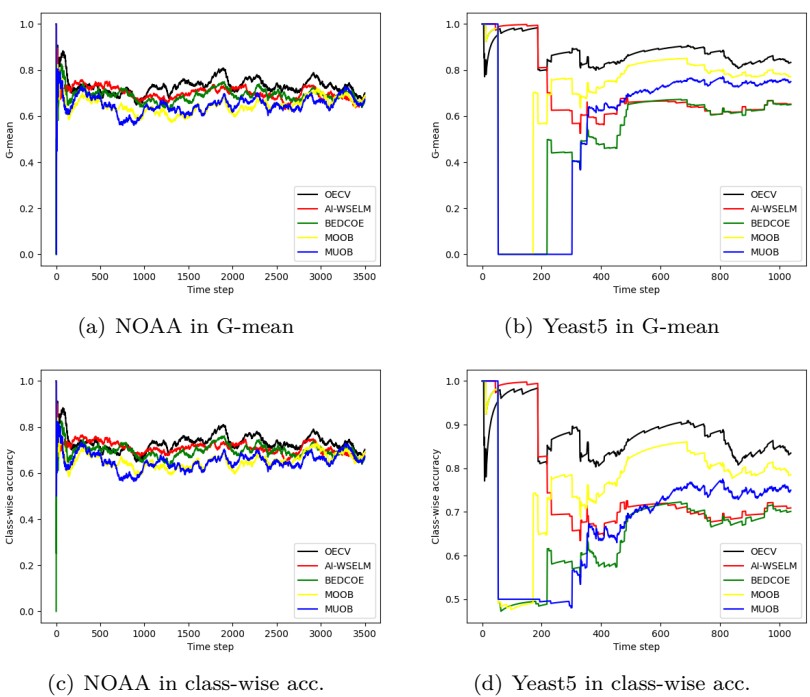

(a) NOAA in G-mean

(b) Yeast5 in G-mean

(c) NOAA in class-wise acc.

(d) Yeast5 in class-wise acc.

Figure 5: Continuous performance comparison throughout time on representative datasets in terms of G-mean and balanced accuracy.

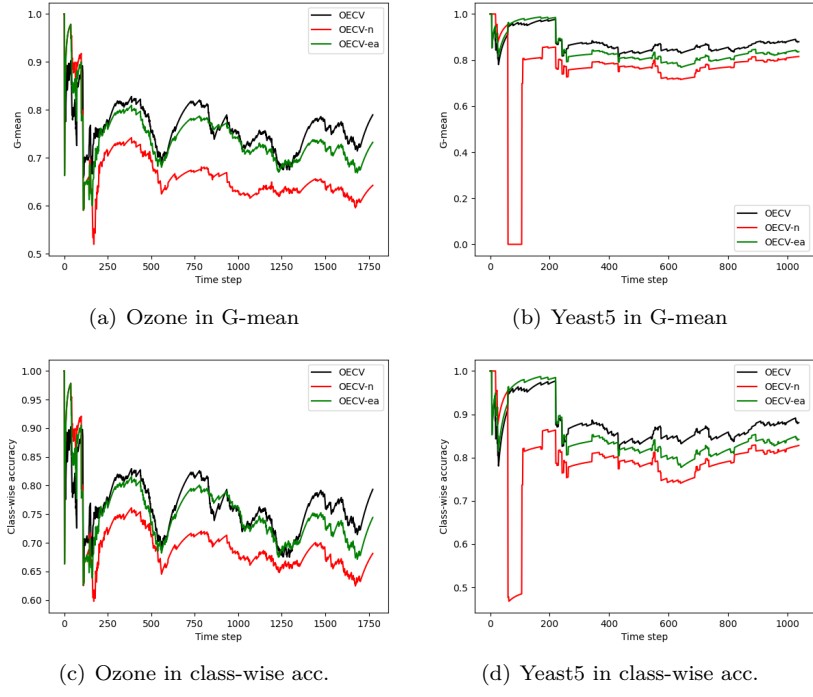

(a) Ozone in G-mean

(b) Yeast5 in G-mean

(c) Ozone in class-wise acc.

(d) Yeast5 in class-wise acc.

Figure 6: Continuous performance comparison of ablation studies throughout time on representative datasets in terms of G-mean and balanced accuracy.

Table 5: Performance comparison between OECV with different population size in terms of G-mean (%) on the left and balanced accuracy (%) on the right. Each entry is the mean±std performance across 10 runs. The best performance on each dataset is highlighted in **bold**, and the 2nd best performance is highlighted in *italics*. The last two rows list the average ranks (avgRank) of each model across datasets, as well as the relative average time costs.

(a) G-mean

| Dataset | Pop-25 | Pop-50 | Pop-100 | Pop-200 |
|---|---|---|---|---|
| Elec | 83.7±7.8 | *83.9±7.8* | 83.8±7.9 | **84.1±7.9** |
| Abrupt | 62.8±3.5 | **63.2±3.5** | *63.1±3.6* | *63.1±3.6* |
| Gradual | 8.5±15.6 | *13.2±19.3* | **24.8±23.6** | 5.5±12.5 |
| Incremental1 | **46.4±5.4** | 46.1±5.7 | 46.2±5.6 | *46.3±5.4* |
| Luxembourg | **100.0±0.0** | **100.0±0.0** | **100.0±0.0** | **100.0±0.0** |
| NOAA | **73.1±4.0** | *73.0±3.9* | 72.9±3.9 | *73.0±4.0* |
| Ozone | 77.1±5.6 | **77.8±5.9** | *77.3±6.0* | 77.0±5.9 |
| Airlines | **51.8±4.7** | 51.7±4.7 | **51.8±4.8** | **51.8±4.7** |
| Covtype | 28.6±15.4 | 36.0±18.4 | *39.0±18.4* | **40.3±18.1** |
| Incremental2 | 15.6±19.5 | 17.2±19.4 | *18.5±17.1* | **20.5±19.3** |
| Abalone1 | *67.8±16.6* | 66.8±16.8 | 67.4±16.7 | **71.9±15.3** |
| Abalone2 | *38.7±24.4* | **51.8±20.2** | 38.2±22.2 | 38.1±20.2 |
| Car1 | *78.2±9.9* | 78.1±9.8 | **78.5±9.9** | 77.6±10.0 |
| Car2 | *96.1±2.0* | **96.2±2.0** | 95.8±1.9 | 95.9±1.7 |
| Kddcup | **100.0±0.0** | 94.9±10.9 | **100.0±0.1** | 98.1±4.5 |
| Kr | *94.7±2.8* | 93.8±3.7 | 93.6±3.7 | **95.1±2.1** |
| Segment | *99.1±0.6* | *99.1±0.6* | **99.2±0.6** | 99.0±0.6 |
| Shuttle1 | **99.9±0.2** | 98.1±6.1 | **99.9±0.2** | 99.8±0.4 |
| Shuttle2 | **99.7±0.6** | 99.6±0.6 | 99.6±0.6 | **99.7±0.6** |
| Thyroid | 71.6±19.2 | 74.4±19.7 | *74.5±19.8* | **76.8±20.3** |
| Win1 | 80.6±18.5 | **84.5±13.8** | 80.3±18.4 | *81.9±15.7* |
| Win2 | 59.2±15.2 | 58.8±12.2 | **61.9±11.3** | *60.1±12.8* |
| Win3 | 79.9±8.2 | **81.5±6.2** | *80.8±6.6* | 80.3±6.6 |
| Win4 | *50.6±20.0* | **64.4±11.9** | 46.6±25.6 | 49.4±25.5 |
| Win5 | 53.3±14.0 | *59.0±9.9* | 57.5±13.3 | **62.8±10.4** |
| Yeast1 | 48.0±29.4 | *51.1±22.6* | 49.2±26.1 | **53.6±22.7** |
| Yeast2 | **0.2±3.0** | *0.1±2.2* | 0.0±0.0 | 0.0±0.9 |
| Yeast3 | **87.8±3.2** | 86.5±3.7 | 86.6±4.0 | *87.2±3.4* |
| Yeast4 | 81.7±13.8 | *86.9±5.6* | 86.5±9.7 | **89.7±4.2** |
| Yeast5 | *86.5±5.8* | **86.6±4.8** | 86.2±5.0 | 85.5±5.0 |
| AvgRank | 2.583 | 2.467 | 2.6 | 2.35 |
| Time cost | ×1 | ×1.11 | ×1.30 | ×1.73 |

(b) Balanced accuracy

| Dataset | Pop-25 | Pop-50 | Pop-100 | Pop-200 |
|---|---|---|---|---|
| Elec | 83.7±7.8 | *83.9±7.8* | 83.8±7.9 | **84.1±7.9** |
| Abrupt | 62.8±3.5 | **63.2±3.5** | *63.1±3.6* | *63.1±3.6* |
| Gradual | 8.5±15.6 | *13.2±19.3* | **24.8±23.6** | 5.5±12.5 |
| Incremental1 | **46.4±5.4** | 46.1±5.7 | 46.2±5.6 | *46.3±5.4* |
| Luxembourg | **100.0±0.0** | **100.0±0.0** | **100.0±0.0** | **100.0±0.0** |
| NOAA | **73.1±4.0** | *73.0±3.9* | 72.9±3.9 | *73.0±4.0* |
| Ozone | 77.1±5.6 | **77.8±5.9** | *77.3±6.0* | 77.0±5.9 |
| Airlines | **51.8±4.7** | 51.7±4.7 | **51.8±4.8** | **51.8±4.7** |
| Covtype | 28.6±15.4 | 36.0±18.4 | *39.0±18.4* | **40.3±18.1** |
| Incremental2 | 15.6±19.5 | 17.2±19.4 | *18.5±17.1* | **20.5±19.3** |
| Abalone1 | *67.8±16.6* | 66.8±16.8 | 67.4±16.7 | **71.9±15.3** |
| Abalone2 | *38.7±24.4* | **51.8±20.2** | 38.2±22.2 | 38.1±20.2 |
| Car1 | *78.2±9.9* | 78.1±9.8 | **78.5±9.9** | 77.6±10.0 |
| Car2 | *96.1±2.0* | **96.2±2.0** | 95.8±1.9 | 95.9±1.7 |
| Kddcup | **100.0±0.0** | 94.9±10.9 | **100.0±0.1** | 98.1±4.5 |
| Kr | *94.7±2.8* | 93.8±3.7 | 93.6±3.7 | **95.1±2.1** |
| Segment | *99.1±0.6* | *99.1±0.6* | **99.2±0.6** | 99.0±0.6 |
| Shuttle1 | **99.9±0.2** | 98.1±6.1 | **99.9±0.2** | 99.8±0.4 |
| Shuttle2 | **99.7±0.6** | 99.6±0.6 | 99.6±0.6 | **99.7±0.6** |
| Thyroid | 71.6±19.2 | 74.4±19.7 | *74.5±19.8* | **76.8±20.3** |
| Win1 | 80.6±18.5 | **84.5±13.8** | 80.3±18.4 | *81.9±15.7* |
| Win2 | 59.2±15.2 | 58.8±12.2 | **61.9±11.3** | *60.1±12.8* |
| Win3 | 79.9±8.2 | **81.5±6.2** | *80.8±6.6* | 80.3±6.6 |
| Win4 | *50.6±20.0* | **64.4±11.9** | 46.6±25.6 | 49.4±25.5 |
| Win5 | 53.3±14.0 | *59.0±9.9* | 57.5±13.3 | **62.8±10.4** |
| Yeast1 | 48.0±29.4 | *51.1±22.6* | 49.2±26.1 | **53.6±22.7** |
| Yeast2 | **0.2±3.0** | *0.1±2.2* | 0.0±0.0 | 0.0±0.9 |
| Yeast3 | **87.8±3.2** | 86.5±3.7 | 86.6±4.0 | *87.2±3.4* |
| Yeast4 | 81.7±13.8 | *86.9±5.6* | 86.5±9.7 | **89.7±4.2** |
| Yeast5 | *86.5±5.8* | **86.6±4.8** | 86.2±5.0 | 85.5±5.0 |
| AvgRank | 2.583 | 2.467 | 2.6 | 2.35 |
| Time cost | ×1 | ×1.11 | ×1.30 | ×1.73 |

individuals) to get four comparison methods: Pop-25 (original setting), Pop-50, Pop-100, Pop-200, standing for OECV with a population size of 25, 50, 100, and 200, respectively. The detailed comparison setting remains the same as in the main paper experiments. We report the performance in terms of G-mean in Table 5 (a) and the performance in terms of balanced accuracy in Table 5 (b).

The result shows that increasing the population size would not boost performance significantly, however, the time complexity increases correspondingly. This can be because the problem scale is commonly small in an online learning setting, meaning a small number of individuals can already find a good enough cost vector. We conclude that OECV is not sensitive to this hyperparameter in a certain range. This is also why we only applied a relatively small population size in our main experiment: this setting can significantly improve performance compared to baseline methods while not incurring a long updating delay. In offline learning, where the problem scale is much larger, especially when the number of classes is larger, a large population size should be applied. We leave the exploration of our method in an offline setting to future work.

Table 6: Performance comparison between OECV with different oversampling rate in terms of G-mean (%) on the left and balanced accuracy (%) on the right. Each entry is the mean±std performance across 10 runs. The best performance on each dataset is highlighted in **bold**, and the 2nd best performance is highlighted in *italics*. The last two rows list the average ranks (avgRank) of each model across datasets, as well as the relative average time costs.

| | (a) G-mean | | | (b) Balanced accuracy | | |
|---|---|---|---|---|---|---|
| Dataset | $r=1$ | $r=3$ | $r=5$ | $r=1$ | $r=3$ | $r=5$ |
| Elec | *83.8±7.4* | **83.9±7.7** | 83.7±7.8 | *84.3±6.5* | **84.4±6.7** | 84.2±6.8 |
| Abrupt | 62.3±3.4 | *62.6±3.6* | **62.8±3.5** | 64.4±2.2 | *64.5±2.2* | **64.6±2.2** |
| Gradual | **8.5±15.5** | 8.5±15.6 | 8.5±15.6 | **52.3±4.7** | 52.3±4.6 | 52.3±4.6 |
| Incremental1 | *46.2±5.6* | *46.2±5.6* | **46.4±5.4** | 47.9±4.3 | *48.0±4.2* | **48.2±4.1** |
| Luxembourg | **100.0±0.0** | **100.0±0.0** | **100.0±0.0** | **100.0±0.0** | **100.0±0.0** | **100.0±0.0** |
| NOAA | 72.8±4.0 | *72.9±3.9* | **73.1±4.0** | *73.0±3.8* | *73.0±3.6* | **73.2±3.8** |
| Ozone | 75.2±5.8 | *76.6±5.8* | **77.1±5.6** | 75.9±5.7 | *76.9±5.8* | **77.4±5.6** |
| Airlines | 51.6±5.0 | *51.7±4.7* | **51.8±4.7** | *52.1±3.9* | *52.1±3.7* | **52.2±3.6** |
| Covtype | 28.2±15.2 | *28.3±15.2* | **28.6±15.4** | **38.7±13.9** | 38.5±13.9 | *38.6±13.9* |
| Incremental2 | *15.7±19.3* | **15.9±19.5** | 15.6±19.5 | **40.1±6.0** | *40.0±5.9* | *40.0±5.9* |
| Abalone1 | 63.0±19.2 | *66.8±16.7* | **67.8±16.6** | 69.5±8.8 | *71.3±8.0* | **71.9±7.9** |
| Abalone2 | 29.8±24.9 | *37.1±24.4* | **38.7±24.4** | 51.9±9.8 | *53.6±9.6* | **54.2±9.8** |
| Car1 | 77.6±9.8 | *78.1±9.9* | **78.2±9.9** | 78.8±7.0 | **79.0±7.1** | **79.0±7.0** |
| Car2 | 89.2±10.9 | **96.7±2.7** | *96.1±2.0* | 90.3±5.9 | **96.8±2.6** | *96.2±2.0* |
| Kddcup | **100.0±0.0** | **100.0±0.0** | **100.0±0.0** | **100.0±0.0** | **100.0±0.0** | **100.0±0.0** |
| Kr | 91.1±3.5 | *93.9±2.7* | **94.7±2.8** | 91.5±3.1 | *94.0±2.6* | **94.7±2.7** |
| Segment | **99.4±0.5** | *99.2±0.6* | 99.1±0.6 | **99.4±0.5** | *99.2±0.6* | 99.1±0.6 |
| Shuttle1 | **99.9±0.1** | 99.9±0.2 | 99.9±0.2 | **99.9±0.1** | 99.9±0.2 | 99.9±0.2 |
| Shuttle2 | **99.7±0.6** | **99.7±0.6** | **99.7±0.6** | **99.7±0.6** | **99.7±0.6** | **99.7±0.6** |
| Thyroid | **72.8±19.4** | *72.0±19.1* | 71.6±19.2 | **76.6±9.0** | *75.5±8.7* | 75.0±9.1 |
| Win1 | **81.0±18.7** | *80.6±18.5* | *80.6±18.5* | **83.9±15.3** | *83.4±15.2* | 83.3±15.2 |
| Win2 | *56.7±15.4* | *56.7±15.3* | **59.2±15.2** | *63.4±7.6* | 63.0±7.5 | **64.6±7.8** |
| Win3 | *80.5±8.4* | **80.8±7.6** | 79.9±8.2 | *81.0±8.4* | **81.2±7.8** | 80.2±8.3 |
| Win4 | 48.8±20.9 | *50.2±20.1* | **50.6±20.0** | **59.6±10.5** | *59.5±10.2* | *59.5±10.2* |
| Win5 | 49.9±14.4 | *53.1±13.9* | **53.3±14.0** | 57.3±10.6 | **58.6±10.2** | *58.5±10.3* |
| Yeast1 | 42.1±31.0 | *47.6±29.1* | **48.0±29.4** | 57.7±17.5 | **59.1±18.0** | *59.0±18.4* |
| Yeast2 | **0.2±3.0** | **0.2±3.0** | **0.2±3.0** | **39.9±4.7** | **39.9±4.7** | **39.9±4.7** |
| Yeast3 | 86.5±3.3 | **87.8±3.6** | **87.8±3.2** | 86.7±2.9 | **87.9±3.1** | **87.9±2.8** |
| Yeast4 | *80.1±13.7* | 79.5±14.7 | **81.7±13.8** | *82.1±7.9* | 81.6±8.3 | **83.4±7.9** |
| Yeast5 | *84.4±8.5* | 83.9±7.8 | **86.5±5.8** | *85.3±7.4* | 84.6±6.9 | **86.8±5.6** |
| avgRank | 2.4 | 1.967 | 1.633 | 2.2 | 2 | 1.8 |
| Time cost | ×1 | ×3.43 | ×4.20 | ×1 | ×3.43 | ×4.20 |

## E.2 Oversampling Rate

We can show the influence of the oversampling rate $r$ in OECV by manually altering only $r$ to get three comparison methods: $r=1$ (i.e., no oversampling), $r=3$, and $r=5$ (original setting). The detailed comparison setting remains the same as in the main paper experiments. We report the performance in terms of G-mean in Table 6 (a) and the performance in terms of balanced accuracy in Table 6 (b).

The result shows that increasing the oversampling rate would boost performance constantly, however, the time complexity also increases. Intuitively, a larger $r$ enhances the sample diversity and allows a more accurate fitness evaluation but makes the fitness evaluation slower. One can use larger $r$ to get further performance improvement, but $r=5$ is good enough to make the fitness evaluation both accurate and efficient.

Table 7: Performance comparison between OECV with different buffer size in terms of G-mean (%) on the left and balanced accuracy (%) on the right. Each entry is the mean±std performance across 10 runs. The best performance on each dataset is highlighted in **bold**, and the 2nd best performance is highlighted in *italics*. The last two rows list the average ranks (avgRank) of each model across datasets, as well as the relative average time costs.

|  | (a) G-mean | | | (b) Balanced accuracy | | |
|---|---|---|---|---|---|---|
| Dataset | $|\mathcal{B}| = 50$ | $|\mathcal{B}| = 100$ | $|\mathcal{B}| = 200$ | $|\mathcal{B}| = 50$ | $|\mathcal{B}| = 100$ | $|\mathcal{B}| = 200$ |
| Elec | **87.6±7.2** | *85.8±7.4* | 83.7±7.8 | **87.8±6.3** | *86.2±6.5* | 84.2±6.8 |
| Abrupt | *62.7±3.2* | 62.3±3.4 | **62.8±3.5** | **64.6±2.3** | 64.2±2.1 | **64.6±2.2** |
| Gradual | 8.2±14.9 | *8.3±15.2* | **8.5±15.6** | *52.1±4.7* | *52.1±4.7* | **52.3±4.6** |
| Incremental1 | *46.0±5.2* | *46.0±5.5* | **46.4±5.4** | *47.7±4.0* | *47.7±4.1* | **48.2±4.1** |
| Luxembourg | **100.0±0.1** | **100.0±0.0** | **100.0±0.0** | **100.0±0.1** | **100.0±0.0** | **100.0±0.0** |
| NOAA | 71.3±3.9 | *72.4±3.8* | **73.1±4.0** | 71.4±3.6 | *72.5±3.5* | **73.2±3.8** |
| Ozone | 74.6±6.1 | *75.6±5.9* | **77.1±5.6** | 75.5±5.8 | *76.2±5.9* | **77.4±5.6** |
| Airlines | *51.8±4.8* | **52.1±4.8** | *51.8±4.7* | *52.2±3.7* | **52.5±3.7** | *52.2±3.6* |
| Covtype | 26.3±14.2 | *27.5±14.8* | **28.6±15.4** | 37.5±13.4 | *37.9±13.5* | **38.6±13.9** |
| Incremental2 | *15.5±19.0* | *15.5±19.3* | **15.6±19.5** | 39.0±5.6 | *39.5±5.8* | **40.0±5.9** |
| Abalone1 | 58.9±18.8 | *59.7±19.6* | **67.8±16.6** | 66.7±7.8 | *67.2±8.6* | **71.9±7.9** |
| Abalone2 | 23.0±22.4 | *29.5±21.4* | **38.7±24.4** | 50.4±7.5 | *50.5±7.0* | **54.2±9.8** |
| Car1 | 75.6±9.1 | *76.9±9.6* | **78.2±9.9** | 77.1±5.9 | *77.9±6.6* | **79.0±7.0** |
| Car2 | **97.5±1.7** | 94.3±3.0 | *96.1±2.0* | **97.5±1.7** | 94.4±3.0 | *96.2±2.0* |
| Kddcup | **100.0±0.0** | **100.0±0.0** | **100.0±0.0** | **100.0±0.0** | **100.0±0.0** | **100.0±0.0** |
| Kr | *93.8±2.8* | 91.0±3.5 | **94.7±2.8** | *93.9±2.6* | 91.2±3.2 | **94.7±2.7** |
| Segment | 98.9±0.7 | **99.1±0.6** | **99.1±0.6** | 98.9±0.7 | **99.1±0.6** | **99.1±0.6** |
| Shuttle1 | **99.9±0.2** | **99.9±0.2** | **99.9±0.2** | **99.9±0.2** | **99.9±0.2** | **99.9±0.2** |
| Shuttle2 | **99.7±0.6** | 99.6±0.6 | **99.7±0.6** | **99.7±0.6** | 99.6±0.6 | **99.7±0.6** |
| Thyroid | 56.3±17.7 | *65.8±19.3* | **71.6±19.2** | 63.7±9.1 | *70.2±11.2* | **75.0±9.1** |
| Win1 | 79.9±19.9 | *80.4±18.9* | **80.6±18.5** | 83.2±15.9 | **83.3±15.4** | **83.3±15.2** |
| Win2 | *57.1±16.2* | 55.2±15.6 | **59.2±15.2** | *64.2±8.0* | 61.9±8.0 | **64.6±7.8** |
| Win3 | 69.8±12.0 | *79.1±8.9* | **79.9±8.2** | 71.4±11.3 | *79.5±9.0* | **80.2±8.3** |
| Win4 | 39.9±24.2 | *50.0±20.8* | **50.6±20.0** | 56.6±10.4 | **60.4±10.0** | *59.5±10.2* |
| Win5 | 52.1±14.9 | *52.2±13.9* | **53.3±14.0** | **59.4±10.3** | 58.1±9.9 | *58.5±10.3* |
| Yeast1 | 41.7±28.1 | *42.3±26.8* | **48.0±29.4** | *58.3±15.8* | 56.6±15.2 | **59.0±18.4** |
| Yeast2 | **0.2±3.0** | **0.2±3.0** | **0.2±3.0** | **39.9±4.7** | **39.9±4.7** | **39.9±4.7** |
| Yeast3 | 84.2±4.3 | *86.0±4.2* | **87.8±3.2** | 84.5±3.8 | *86.2±3.8* | **87.9±2.8** |
| Yeast4 | 76.1±17.6 | *77.5±16.1* | **81.7±13.8** | 79.5±8.9 | *80.3±8.4* | **83.4±7.9** |
| Yeast5 | 82.0±7.9 | *85.0±6.6* | **86.5±5.8** | 82.9±7.4 | *85.4±6.4* | **86.8±5.6** |
| avgRank | 2.533 | 2.15 | 1.317 | 2.417 | 2.167 | 1.417 |
| Time cost | ×1 | ×1.28 | ×1.77 | ×1 | ×1.28 | ×1.77 |

### E.3 Buffer Size

We can show the influence of the buffer size $|\mathcal{B}|$ in OECV by manually altering only the buffer size to get three comparison methods: $|\mathcal{B}| = 50$, $|\mathcal{B}| = 100$, and $|\mathcal{B}| = 200$ (original setting). The detailed comparison setting remains the same as in the main paper experiments. We report the performance in terms of G-mean in Table 7 (a) and the performance in terms of balanced accuracy in Table 7 (b).

The result shows that increasing the buffer size would boost performance constantly, however, both the time complexity and storage complexity increase. Intuitively, a larger buffer makes more samples available to the cost vector and allows a more accurate fitness evaluation but makes the fitness evaluation slower. One can use a larger buffer to get further performance improvement, but $|\mathcal{B}| = 200$ is good enough to make the fitness evaluation both accurate and efficient.

Table 8: Performance comparison between OECV with different updating frequency of cost vector in terms of G-mean (%) on the left and balanced accuracy (%) on the right. Each entry is the mean±std performance across 10 runs. The best performance on each dataset is highlighted in **bold**, and the 2nd best performance is highlighted in *italics*. The last two rows list the average ranks (avgRank) of each model across datasets, as well as the relative average time costs.

| | (a) G-mean | | | (b) Balanced accuracy | | |
|---|---|---|---|---|---|---|
| Dataset | $f = 5$ | $f = 10$ | $f = 20$ | $f = 5$ | $f = 10$ | $f = 20$ |
| Elec | **83.7±7.8** | *83.6±7.7* | *83.6±7.6* | **84.2±6.8** | *84.1±6.7* | *84.1±6.6* |
| Abrupt | **62.8±3.5** | **62.8±4.1** | 62.6±4.6 | **64.6±2.2** | **64.6±2.0** | 64.5±2.1 |
| Gradual | *8.5±15.6* | 4.5±8.1 | **11.0±13.6** | **52.3±4.6** | *50.7±4.3* | 43.1±6.6 |
| Incremental1 | **46.4±5.4** | *46.1±5.5* | *46.1±5.7* | **48.2±4.1** | *48.0±4.0* | 47.9±4.1 |
| Luxembourg | **100.0±0.0** | **100.0±0.0** | **100.0±0.0** | **100.0±0.0** | **100.0±0.0** | **100.0±0.0** |
| NOAA | **73.1±4.0** | *72.9±4.1* | 72.7±4.0 | **73.2±3.8** | *73.0±3.8* | 72.8±3.7 |
| Ozone | **77.1±5.6** | **77.1±5.8** | 76.8±5.8 | **77.4±5.6** | *77.3±5.8* | 77.1±5.8 |
| Airlines | **51.8±4.7** | **51.8±4.7** | 51.7±4.8 | **52.2±3.6** | **52.2±3.6** | 52.1±3.7 |
| Covtype | *28.6±15.4* | 28.4±15.7 | **37.8±18.4** | *38.6±13.9* | 37.5±13.5 | **46.4±8.4** |
| Incremental2 | **15.6±19.5** | 12.4±15.3 | *13.5±16.6* | **40.0±5.9** | *28.6±8.6* | 27.4±10.2 |
| Abalone1 | *67.8±16.6* | **69.7±14.9** | 65.4±18.4 | *71.9±7.9* | **72.9±7.8** | 69.8±9.1 |
| Abalone2 | 38.7±24.4 | *39.9±22.4* | **50.7±20.7** | *54.2±9.8* | 53.6±8.8 | **55.6±14.5** |
| Car1 | **78.2±9.9** | *77.4±9.9* | 76.6±10.1 | **79.0±7.0** | *78.3±7.1* | 77.5±7.5 |
| Car2 | 96.1±2.0 | *96.3±1.7* | **97.1±1.0** | 96.2±2.0 | *96.4±1.7* | **97.2±1.0** |
| Kddcup | **100.0±0.0** | **100.0±0.0** | 95.6±9.7 | **100.0±0.0** | **100.0±0.0** | 96.1±8.1 |
| Kr | 94.7±2.8 | **96.1±2.3** | *95.5±1.9* | 94.7±2.7 | **96.1±2.2** | *95.5±1.9* |
| Segment | **99.1±0.6** | **99.1±0.6** | **99.1±0.5** | **99.1±0.6** | **99.1±0.6** | **99.1±0.5** |
| Shuttle1 | **99.9±0.2** | **99.9±0.2** | 98.4±5.1 | **99.9±0.2** | **99.9±0.2** | 98.5±4.4 |
| Shuttle2 | **99.7±0.6** | **99.7±0.6** | 99.6±0.6 | **99.7±0.6** | **99.7±0.6** | 99.6±0.6 |
| Thyroid | 71.6±19.2 | *72.8±19.5* | **72.9±19.6** | 75.0±9.1 | **76.2±9.5** | **76.2±9.7** |
| Win1 | **80.6±18.5** | *79.9±18.6* | 74.4±24.8 | **83.3±15.2** | *82.7±15.2* | 75.6±23.4 |
| Win2 | **59.2±15.2** | **59.2±15.2** | 57.7±15.0 | **64.6±7.8** | *64.2±7.8* | 63.3±7.7 |
| Win3 | 79.9±8.2 | *80.6±7.1* | **81.3±6.9** | 80.2±8.3 | *81.0±7.3* | **81.7±7.1** |
| Win4 | *50.6±20.0* | 46.7±20.3 | **63.7±15.9** | *59.5±10.2* | 56.3±9.3 | **67.2±10.9** |
| Win5 | **53.3±14.0** | 49.8±14.6 | *50.3±14.5* | **58.5±10.3** | 56.2±9.5 | *56.8±9.5* |
| Yeast1 | **48.0±29.4** | *46.6±25.3* | 46.0±24.2 | **59.0±18.4** | *57.0±16.6* | 55.1±17.3 |
| Yeast2 | **0.2±3.0** | *0.0±1.6* | *0.0±1.4* | **39.9±4.7** | *30.0±5.0* | 28.6±5.9 |
| Yeast3 | **87.8±3.2** | *87.2±3.5* | 87.1±3.3 | **87.9±2.8** | *87.3±3.1* | 87.2±2.8 |
| Yeast4 | 81.7±13.8 | **87.2±9.5** | *86.6±5.5* | 83.4±7.9 | **87.8±5.9** | *86.9±5.2* |
| Yeast5 | **86.5±5.8** | *86.1±5.2* | 83.4±6.6 | **86.8±5.6** | *86.3±5.1* | 84.0±6.2 |
| avgRank | 1.717 | 2.0 | 2.283 | 1.617 | 1.95 | 2.433 |
| Time cost | ×2.57 | ×1.36 | ×1 | ×2.57 | ×1.36 | ×1 |

### E.4 Updating Frequency

We can show the influence of the updating frequency $f$ of the cost vector in OECV by manually altering only the $f$ to get three comparison methods: $f = 5$ (original setting), $f = 10$, and $f = 20$. The detailed comparison setting remains the same as in the main paper experiments. We report the performance in terms of G-mean in Table 8 (a) and the performance in terms of balanced accuracy in Table 8 (b).

The result shows that decreasing the update would boost performance constantly, however, the time complexity increases. Intuitively, a smaller $f$ makes the updating frequency more aligned with the classifier in the lower layer. This reduces the probability of updating delay and a sub-optimal solution. One can use a smaller $f$ to get further performance improvement, but $f = 5$ is good enough, and we pick this value to save the runtime of OECV.

Table 9: Performance comparison between OECV with different pretrain ratio in terms of G-mean (%) on the left and balanced accuracy (%) on the right. Each entry is the mean±std performance across 10 runs. The best performance on each dataset is highlighted in **bold**, and the 2nd best performance is highlighted in *italics*. The last two rows list the average ranks (avgRank) of each model across datasets, as well as the relative average time costs.

(a) G-mean

| Dataset | $Ratio = 0$ | $Ratio = 0.1$ | $Ratio = 0.3$ |
|---|---|---|---|
| Elec | **84.7±6.9** | 83.3±7.1 | *83.7±7.8* |
| Abrupt | **65.4±9.6** | 61.8±8.2 | *62.8±3.5* |
| Gradual | **14.7±25.2** | 7.9±15.4 | *8.5±15.6* |
| Incremental1 | **51.1±9.7** | *48.2±5.4* | 46.4±5.4 |
| Luxembourg | 95.2±11.0 | *97.9±4.0* | **100.0±0.0** |
| NOAA | 63.1±5.7 | *69.5±4.1* | **73.1±4.0** |
| Ozone | **77.2±9.1** | 75.3±9.9 | *77.1±5.6* |
| Airlines | *50.9±4.3* | 49.9±3.7 | **51.8±4.7** |
| Covtype | *27.2±23.8* | 25.7±20.5 | **28.6±15.4** |
| Incremental2 | **19.7±23.2** | 14.2±20.5 | *15.6±19.5* |
| Abalone1 | **72.9±10.6** | 67.1±9.8 | *67.8±16.6* |
| Abalone2 | *46.7±15.4* | **51.8±12.2** | 38.7±24.4 |
| Car1 | 56.7±7.5 | *71.5±7.1* | **78.2±9.9** |
| Car2 | 77.7±14.7 | *92.8±2.5* | **96.1±2.0** |
| Kddcup | *98.9±2.5* | 97.4±10.1 | **100.0±0.0** |
| Kr | 91.5±10.7 | *93.9±3.9* | **94.7±2.8** |
| Segment | 93.4±10.3 | *98.8±0.6* | **99.1±0.6** |
| Shuttle1 | 98.3±5.7 | *99.2±0.9* | **99.9±0.2** |
| Shuttle2 | **99.7±0.6** | 99.6±0.5 | **99.7±0.6** |
| Thyroid | 51.7±16.0 | *54.3±21.3* | **71.6±19.2** |
| Win1 | 71.1±18.9 | **88.2±11.9** | *80.6±18.5* |
| Win2 | **62.2±11.3** | 49.8±17.5 | *59.2±15.2* |
| Win3 | *62.8±20.7* | 27.6±28.8 | **79.9±8.2** |
| Win4 | *46.5±23.5* | 37.8±29.9 | **50.6±20.0** |
| Win5 | **60.4±13.4** | 29.9±29.0 | *53.3±14.0* |
| Yeast1 | **64.1±9.6** | 42.6±16.8 | *48.0±29.4* |
| Yeast2 | **5.1±9.1** | *0.7±7.0* | 0.2±3.0 |
| Yeast3 | *86.0±8.1* | 85.2±6.5 | **87.8±3.2** |
| Yeast4 | *92.2±8.8* | **93.4±2.9** | 81.7±13.8 |
| Yeast5 | *77.5±18.5* | 50.7±32.3 | **86.5±5.8** |
| avgRank | 1.917 | 2.467 | 1.617 |
| Time cost | ×1.27 | ×1.17 | ×1 |

(b) Balanced accuracy

| Dataset | $Ratio = 0$ | $Ratio = 0.1$ | $Ratio = 0.3$ |
|---|---|---|---|
| Elec | **85.0±6.5** | 83.8±6.7 | *84.2±6.8* |
| Abrupt | **67.0±7.6** | 64.4±3.1 | *64.6±2.2* |
| Gradual | **52.6±10.8** | 51.3±4.7 | *52.3±4.6* |
| Incremental1 | **53.0±8.8** | *49.6±4.6* | 48.2±4.1 |
| Luxembourg | 95.4±10.2 | *97.9±3.8* | **100.0±0.0** |
| NOAA | 63.7±4.6 | *69.7±3.7* | **73.2±3.8** |
| Ozone | **77.6±8.0** | 76.2±7.2 | *77.4±5.6* |
| Airlines | *51.2±3.7* | 50.3±3.4 | **52.2±3.6** |
| Covtype | *35.8±26.8* | 35.7±22.8 | **38.6±13.9** |
| Incremental2 | *40.9±10.6* | **41.3±7.2** | 40.0±5.9 |
| Abalone1 | **73.4±9.8** | 69.0±5.6 | *71.9±7.9* |
| Abalone2 | 51.9±9.1 | **54.9±7.6** | *54.2±9.8* |
| Car1 | 57.2±7.1 | *72.1±6.6* | **79.0±7.0** |
| Car2 | 78.8±10.9 | *92.9±2.5* | **96.2±2.0** |
| Kddcup | *98.9±2.4* | 97.9±5.9 | **100.0±0.0** |
| Kr | 91.7±10.1 | *94.0±3.5* | **94.7±2.7** |
| Segment | 93.7±9.3 | *98.8±0.6* | **99.1±0.6** |
| Shuttle1 | 98.4±5.3 | *99.2±0.9* | **99.9±0.2** |
| Shuttle2 | **99.7±0.6** | 99.6±0.5 | **99.7±0.6** |
| Thyroid | 55.7±9.7 | *64.7±7.3* | **75.0±9.1** |
| Win1 | 74.4±12.2 | **88.8±11.0** | *83.3±15.2* |
| Win2 | *63.3±10.5* | 55.4±7.1 | **64.6±7.8** |
| Win3 | *64.4±19.0* | 53.9±12.4 | **80.2±8.3** |
| Win4 | 55.6±15.8 | *57.6±13.2* | **59.5±10.2** |
| Win5 | **62.4±12.7** | 51.7±16.1 | *58.5±10.3* |
| Yeast1 | **65.4±8.6** | 50.8±5.9 | *59.0±18.4* |
| Yeast2 | 25.4±10.5 | *39.0±5.3* | **39.9±4.7** |
| Yeast3 | *86.3±7.4* | 85.4±6.0 | **87.9±2.8** |
| Yeast4 | *92.5±8.1* | **93.5±2.9** | 83.4±7.9 |
| Yeast5 | *78.2±16.9* | 65.2±13.7 | **86.8±5.6** |
| avgRank | 2.117 | 2.367 | 1.517 |
| Time cost | ×1.27 | ×1.17 | ×1 |

### E.5 Pre-training Ratio

We can show the influence of the ratio of the dataset for pretraining in OECV by manually altering only the pretraining ratio to get three comparison methods: $Ratio = 0$ (begin from scratch), $Ratio = 0.1$, and $Ratio = 0.3$ (original setting). The detailed comparison setting remains the same as in the main paper experiments. Note the model is evaluated only on the remaining stream after the pretraining stage. We report the performance in terms of G-mean in Table 9 (a) and the performance in terms of balanced accuracy in Table 9 (b).

The result does not show an obvious relation between the pretraining ratio and performance in our method. Indeed, this hyperparameter is not an essential part of our method, and OECV can start from scratch ($Ratio = 0$). The hyperparameter is retained to align with the compared method Li et al. (2023), and choosing a proper ratio and setting it equally to all compared methods is enough to make the comparison fair, as we did in the main experiment.

# F   More Experimental Comparison with Comparable Storage Budget

Except for the baseline AI-WSELM (Qin et al., 2021), which also requires extra storage as the same as ours, the other three baselines MOOB, MUOB (Wang et al., 2016) and BEDCOE (Li et al., 2023), do not have this requirement. In this section, We compare with an additional baseline named Online SMOTE Bagging (SmoteOB) (Wang & Pineau, 2016) that also uses extra storage to demonstrate the superiority of OECV when the compared method enjoys comparable or even higher storage requirements. The SmoteOB oversamples using training samples within a sliding window, and we set the size of the sliding window to 100 for each class (i.e., at least 200 samples to be stored for all classes), being equal to or larger than ours.

We report the performance in terms of G-mean in Table 10 (a) and the performance in terms of balanced accuracy in Table 10 (b). We can draw the observation that OECV outperforms SmoteOB with a similar time cost. An analysis analogous to the main paper can explain that our method performs better in cases where few classes are presented, and the stream is highly imbalanced. This illustrates that our method can not only outperform baselines with no extra storage requirement but also outperform baselines with extra storage used, verifying the effectiveness of OECV.

Table 10: Performance comparison between OECV and SmoteOB in terms of G-mean (%) on the left and balanced accuracy (%) on the right. Each entry is the mean±std performance across 10 runs. The best performance on each dataset is highlighted in **bold**, and the 2nd best performance is highlighted in *italics*. The last two rows list the average ranks (avgRank) of each model across datasets, as well as the relative average time costs.

|              | (a) G-mean     |                  | (b) Balanced accuracy |                  |
|--------------|----------------|------------------|-----------------------|------------------|
| Dataset      | OECV           | SmoteOB          | OECV                  | SmoteOB          |
| Elec         | 83.7±0.9       | **87.5 ± 0.4**   | 84.2±0.9              | **87.9 ± 0.5**   |
| Abrupt       | **62.8±0.6**   | 51.4 ± 13.0      | **64.6±0.6**          | 62.5 ± 0.9       |
| Gradual      | 8.5±4.2        | **8.7 ± 4.6**    | **52.3±1.0**          | 47.4 ± 1.1       |
| Incremental1 | 46.4±1.5       | **51.4 ± 0.7**   | 48.2±1.3              | **55.4 ± 0.7**   |
| Luxembourg   | **100.0±0.0**  | 99.6 ± 0.2       | **100.0±0.0**         | 99.6 ± 0.2       |
| NOAA         | **73.1±0.5**   | 64.6 ± 0.3       | **73.2±0.5**          | 66.1 ± 0.3       |
| Ozone        | 77.1±1.7       | **77.7 ± 1.7**   | 77.4±1.6              | **78.4 ± 1.4**   |
| Airlines     | **51.8±0.9**   | 35.6 ± 1.9       | **52.2±0.8**          | 49.2 ± 0.5       |
| Covtype      | 28.6±1.5       | **51.0 ± 3.5**   | 38.6±1.1              | **63.3 ± 1.6**   |
| Incremental2 | **15.6±1.6**   | 4.8 ± 1.2        | 40.0±0.6              | **43.3 ± 0.8**   |
| Abalone1     | **67.8±4.3**   | 57.1 ± 1.7       | **71.9±2.5**          | 62.8 ± 1.4       |
| Abalone2     | 38.7±7.6       | **41.0 ± 1.9**   | 54.2±3.1              | **57.0 ± 0.9**   |
| Car1         | 78.2±2.2       | **89.8 ± 1.9**   | 79.0±2.1              | **90.2 ± 1.8**   |
| Car2         | **96.1±1.0**   | 78.8 ± 2.7       | **96.2±1.0**          | 81.7 ± 2.1       |
| Kddcup       | **100.0±0.0**  | 71.9 ± 0.7       | **100.0±0.0**         | 78.8 ± 0.6       |
| Kr           | **94.7±1.3**   | 88.2 ± 1.6       | **94.7±1.2**          | 89.4 ± 1.4       |
| Segment      | **99.1±0.1**   | 95.3 ± 0.5       | **99.1±0.1**          | 95.4 ± 0.5       |
| Shuttle1     | **99.9±0.0**   | **99.9 ± 0.1**   | **99.9±0.0**          | **99.9 ± 0.1**   |
| Shuttle2     | **99.7±0.0**   | 99.5 ± 0.1       | **99.7±0.0**          | 99.5 ± 0.1       |
| Thyroid      | **71.6±1.5**   | 52.1 ± 0.5       | **75.0±1.5**          | 59.8 ± 0.4       |
| Win1         | **80.6±1.2**   | 70.4 ± 24.3      | **83.3±0.5**          | 79.5 ± 10.3      |
| Win2         | **59.2±3.6**   | 51.7 ± 0.6       | **64.6±2.0**          | 60.8 ± 0.4       |
| Win3         | **79.9±1.2**   | 62.5 ± 1.0       | **80.2±1.2**          | 66.3 ± 0.7       |
| Win4         | 50.6±5.7       | **68.3 ± 2.9**   | 59.5±4.4              | **72.2 ± 2.1**   |
| Win5         | 53.3±7.1       | **78.7 ± 0.4**   | 58.5±3.8              | **80.0 ± 0.3**   |
| Yeast1       | 48.0±18.9      | **55.5 ± 1.3**   | **59.0±7.7**          | 58.3 ± 1.0       |
| Yeast2       | 0.2±0.4        | **9.8 ± 7.4**    | 39.9±1.1              | **44.1 ± 1.2**   |
| Yeast3       | **87.8±1.0**   | 84.7 ± 0.5       | **87.9±1.0**          | 86.6 ± 0.5       |
| Yeast4       | 81.7±5.7       | **93.1 ± 0.2**   | 83.4±4.3              | **93.2 ± 0.2**   |
| Yeast5       | **86.5±1.7**   | 83.0 ± 1.0       | **86.8±1.6**          | 83.6 ± 0.9       |
| avgRank      | 1.42           | 1.58             | 1.38                  | 1.62             |
| Time cost    | ×1.32          | ×1               | ×1.32                 | ×1               |

