# OpenReview forum: "Threshold Moving for Online Class Imbalance Learning with Dynamic Evolutionary Cost Vector"
_TMLR — Accepted by TMLR_

### Review · Reviewer_MURu · 2024-07-13

**Summary Of Contributions:**

The paper introduces a novel online learning framework that can handle class imbalance and concept drift by updating a cost vector using evolutionary algorithms. The authors model the online learning task as a bi-level optimization problem where the lower-level optimization involves minimizing a normal classification loss while the upper-level optimization involves learning a cost vector using evolutionary algorithms to correct classifier predictions in the presence of data drift. The algorithm is empirically validated on 30 real world datasets and compared with 4 SOTA baselines.

**Audience:**

Yes

**Claims And Evidence:**

Yes

**Requested Changes:**

1. Please comment on the weaknesses above. I do not see them as dealbreakers but if my concerns are valid I would like to see them at least discussed in the paper.

2. Consider adding examples of the kind of online performance metrics (especially non-differentiable ones) that could be considered, in paragraph 2 of the introduction.

3. Can you explain how what is "ad-hoc" about the heuristics chosen by the related works cited in Section 2.1.1 and why these could hinder the performance?

4. The cost-vector based correction for class imbalance appears to assume that the classifier is well-calibrated (misclassified samples will be corrected if their predicted class probabilities are low). Will this actually be the case in online settings where the classifier is trained with a small number of samples and so may become overconfident? Can any calibration approaches (like Platt scaling) be used to address this issue?

5. There is a reference to Line 16 in Algorithm 2 in Section 3.4.1 but the algorithm block only has 15 lines. Please fix this.

6.While the comparisons between OECV-ea and OECV as well as OECV-n and OECV-ea seem to suggest that extra data does not play a determinative role, it isn't clear if it is at all useful. Will it be possible to add a version which uses the evolutionary algorithm but does not use the extra data to see if having the extra data makes any material difference?

7.Why is high correlation between standard deviation and class imbalance a good thing?

**Strengths And Weaknesses:**

Strengths:

1. The use of evolutionary algorithms to cast the class imbalance problem as a hyperparameter optimization problem is an interesting and novel approach that can be applied to a wide range of performance metrics

2. The experimental results show that the approach generally outperforms the SOTA baselines and the ablations studies show that the gains are indeed due to the use of evolutionary algorithms.

Weaknesses:

1. The mismatch in the update frequencies of the upper and lower-level optimization problems may lead to sub-optimal choices of the cost vector (since the classifier is updated more frequently).

2. The oversampling approach appears to be a) susceptible to overfitting if the same samples from the buffer $\mathcal{B}$ are used to train the classifier and then added to $\mathcal{B'}$ to update the cost vector, and b) susceptible to noise in the class labels since the effect of erroneous labels will be amplified by the generated samples in (5) (if the original sample is corrupted then all the generated samples will also be corrupted since they have the same label).

---

> ### Author Response · Authors · 2024-08-04
> **Response to Reviewer MURu**
>
> We would like to thank the reviewer for the very detailed feedback and the constructive comments.
>
> *Weaknesses 1* We agree that updating the upper level less frequently may cause a sub-optimal solution. The reason for introducing frequency $f$ is mainly because of time consumption considerations. One can set $f=1$ if computation resources are sufficient, in which case the updating frequency of two levels align, and there is less chance of incurring a sub-optimal solution. We have also added an experiment on the influence of $f$ in our method. Please refer to Appendix E.4 for a discussion.
>
> *Weaknesses 2*
>
> For *Weaknesses 2 a)*, if in offline learning, a better choice is to split the dataset into non-overlapping training and validation datasets to train the two levels. However, this is more tricky in our online setting since the samples come in the form of a stream. We didn't add additional design for simplicity. In fact, the oversampling technique, which, although not originally proposed to handle this problem but proposed to enhance the optimization of the cost vector, may also help.  Specifically, this can alleviate the overfitting of overlapping data sources by introducing diversified data via interpolation. That is, the data used for optimizing the cost vector has a difference compared to that used for the classifier.
>
> For *Weaknesses 2 b)* We agree that this situation exists. However, we haven't considered handling label noise in our current work since it's not the focus of our topic, and we didn't add additional design. However, it's still a valuable point to alleviate the impact of corrupted labels for future research.
>
> *Requested Changes 1,* We also added a Remark1 in Section 3.2 to discuss the potential issue of the overfitting issue in response to *Weakness 2 a)*. A discussion of the issues of label noise is added as a future work of the conclusion section in response to *Weakness 2 b)*.
>
> *Requested Changes 2* In fact, a wide range of metrics are non-differentiable since they depend on the confusion matrix, which involves a non-differentiable step loss function. We have added more details on this point in paragraph 2 of the introduction.
>
> *Requested Changes 3* We apologize for being misleading on this point. We want to emphasize that all three kinds of online class imbalance algorithms (including data-level, algorithm-level, and ensemble method) suffer from ad hoc heuristics.  To further illustrate this point and motivate our method, we provide more details in a separate section, 2.1.4.
>
> *Requested Changes 4* This is a valuable question. Classifier calibration in online learning remains unexplored. Offline, one can use well-studied techniques such as Platt scaling, matrix and vector scaling, and temperature scaling [1]. However, they cannot be directly adapted to our online setting since an extra calibration set or validation set is usually required, which is not available in online learning. We leave out how to incorporate online classifier calibration into our framework for future work.
>
> *Requested Changes 5* We corrected: It should refer to Line 15.
>
> *Requested Changes 6* We apologize for the potential confusion here. In our current method, a dataset (i.e., buffer $\mathcal B$) has to be created in order to evaluate the cost vector because we cannot calculate the fitness on a single data point. Compared to other baseline methods that do not save any samples processed before, our method has to save them in the buffer, which may cause an unfair comparison. The point of making the baseline OECV-ea is that we want to verify that the buffered data is not the determinative reason why our method gains outperformance.
>
> *Requested Changes 7* We apologize for the mistake here. What should be compared to the population standard deviation is the **change ratio** of the class imbalance ratio instead of the class imbalance ratio itself. The change ratio represents the degree of changes, which should be positively correlated to the population standard deviation to give the conclusion. We have changed the figure in Section 4.4.1, and the conclusion still holds. This highlights that OECV can dynamically change the population diversity as the imbalance ratio changes. Please see Section 4.4.1 for detailed discussion.
>
> Thank you again for engaging with us and for your valuable feedback.
>
> [1] Guo, Chuan, et al. "On calibration of modern neural networks." *International conference on machine learning*. PMLR, 2017.

---

### Review · Reviewer_rcK9 · 2024-07-31

**Summary Of Contributions:**

The paper proposes Online Evolutionary Cost Vector (OECV), a method that aims at handling imbalaced scenarios in online learning using evolutionary algorithm (EA). The paper proposes a dynamic EA to adaptively learn the cost vector in the presence of concept drift, which unifies EA and online class imbalance learning within a bi-level optimization framework by applying a cost vector. The first layer, an online classifier, offers a rough probabilistic prediction and updates by its own rule. The second layer, a cost vector, refines the rough prediction and undergoes a dynamic optimization process via dynamic EA. OECV performance is tested on 30 real-world datasets, in comparison to four state-of-the-art (SOTA) methods.

**Audience:**

Yes

**Broader Impact Concerns:**

No broader impact concerns.

**Claims And Evidence:**

No

**Requested Changes:**

- Please add ablation studies for hyperparameters such as the samples used for initilization, buffer size and oversampling rate, and make some conclusions related to the impact of those on the performance of your method
- Please justify all your technical decisions (such as those related to the hyperparameters mentioned in the previous item) as you present your main experiments
- Please provide results on the computational cost of the proposed method compared to others

**Strengths And Weaknesses:**

Strengths:
- This paper studies an important problem of online imbalanced learning, and proposes a reasonable mechanism to address this problem that relies on bi-level optimization using EA
- The paper is in general well-written, structured and presented

Weaknesses:
- In the experiments, there seem to be some arbitrary decisions that are loosely justified. For example, the initial 30% samples of each stream are used for model initialization in an offline fashion. How is this rate decided? More importantly, what is the impact of this initial sample size and 'pre-training' mechanism in the overall effectiveness of the proposed method? The same question applied to other decisions such as the buffer size and the oversampling rate. Are these data-dependent, method-dependent? What is the impact these would have on the performance of OECV?
- Some ablation studies seem to be missing, in particular those related to the hyper-parameters mentioned in the previous item
- In the experiments, no analysis are provided in terms of the computational cost (memory or computation), with respect to other SOTA methods. How expensive, computationally, would running the proposed OECV be? This is extremely important considering the well-known cost of EA algorithms.

---

> ### Author Response · Authors · 2024-08-04
> **Response to Reviewer rcK9**
>
> We would like to thank the reviewer for the very detailed feedback and the constructive comments.
>
> *Weaknesses*: Please refer to the following Requested Changes.
>
> *Requested Changes 1-2*
>
> Besides the analysis on the population size that is already included, we have added experiment and corresponding analysis on four hyperparameters, namely the oversampling rate $r$, buffer size $|\mathcal{B}|$, the updating frequency of cost vector $f$, and the ratio of the stream for pretraining. Please refer to Remark 2 in Section 3.2 for an intuitive explanation and Appendix E for empirical experiments.
>
> Most of the hyperparameters in our work are raised to handle the performance-resource trade-off, including oversampling rate and buffer size. In practice, we keep the oversampling rate and buffer size as small as possible and do not do further fine-tuning. The goal is to align with real-world problems where time, computational resources, and memory are limited.
>
> We want to clarify that the ratio for pretraining is not necessary in our method. The reason why we still apply the pretraining is to make a fair comparison since one of the compared baselines [1] requires a setup of offline training. We randomly selected and didn't fine-tune the ratio of $30\%$ in the experiment. We also provide an analysis of the effect of the pretraining ratio in Appendix E.5.
>
> We want to emphasize that although our method includes amount of hyperparameters, they are relatively robust in a certain range. We keep the same set of hyperparameters for experiments on 30 datasets, where our method outperforms or is on par with baseline methods. This demonstrates that only a little effort has to be put into choosing the right hyperparameters.
>
> *Requested Changes 3*
>
> All the compared baselines and our methods are of a time complexity of $O(T)$, i.e., being linear to the length of the stream, while the difference lies in the cost at each time step. Besides the empirical result in Appendix A showing the practicality of our method, we also provided a formal analysis of the computational cost of OECV in Section 3.4.4 to give theoretical evidence on this point.
>
> Thank you again for engaging with us and for your valuable feedback.
>
>
>
> [1] Li, Shuxian, et al. "BEDCOE: Borderline Enhanced Disjunct Cluster Based Oversampling Ensemble for Online Multi-Class Imbalance Learning." *ECAI 2023: 26th European Conference on Artificial Intelligence, September 30–October 4, 2023, Kraków, Poland, Including 12th Conference on Prestigious Applications of Intelligent Systems (PAIS 2023): Proceedings*. IOS Press BV, 2023.

---

### Review · Reviewer_gCJ4 · 2024-08-01

**Summary Of Contributions:**

This paper addresses online class imbalance learning. The proposed approach employes the threshold moving method (Section 2), where the output of the model prediction is modified by a predefined weights in the standard approaches. In the proposed approach, this weights are optimized during the training by utilizing an evolutionary approaches. The proposed approach works as follows. The baseline classifier is trained independently of the weighting scheme, by using online stream data. The weight vector is optimized so that the validation performance is maximized, where the validation performance is measured by keeping online data in a buffer and evaluating the performance on them. The proposed approach is compared with several baselines on 30 datasets. The reported results show its promising performance over the compared approaches.

**Audience:**

Yes

**Broader Impact Concerns:**

no concern

**Claims And Evidence:**

No

**Requested Changes:**

P3 “ These methods heavily rely on ad hoc heuristics and hyperparameters, like safety degree (Jiang et al., 2021), sampling rate (Wang & Pineau, 2016), borderline factor, and disjunct factor (Ren et al., 2019), which potentially hinder optimal performance.” As I am not familiar with these approaches, I could not get from what is written how complicated it is to choose a reasonable parameter. As this is one of the main motivation of the proposed approach, please elaborate on it.

P5 “The overall bi-level optimization problem is stated as  …” It is wrong. The lower level loss is defined by ell_1, whereas the upper level is defined by ell_2.

Moreover, although the authors say that the resulting problem is a bi-level optimization problem, as the lower level optimization is independent of the upper level, it looks like just a single layer optimization with a dynamic objective where the objective is changing due to the training of the classifier.

P6 “In other words, the model faces heavier penalties for misclassifying class i to class j as class j gets larger or class i gets smaller. ” I am confused here. If I understand correctly, the cost function is untouched and only the output of the model is modified by M.

P8 “The buffer size |B| for OECV is fixed at 200 samples, and the oversampling rate is set to 3 for all datasets” It is contradicting the setting of online learning from streaming data described in Section 3.1. 200 out of the dataset size of at most 5000 (less than 2000 for for half of the datasets tested in this paper)  is definitely not a small buffer. And it is even enlarged by the factor of 3, which already amount to the size of the whole dataset of several test datasets. If one can have such a huge buffer, why not use thse buffer to learn a lower level classifier? Isn’t it possible to easily approximate the approaches for offline class imbalance learning by using such a buffer? Do the compared approaches listed in Section 4.1 use the same amount of memory? Please compare the memory usage and execution time as well as the performance.

P9 How are the datasets splited to the training and test datasets? How many data for each? Doesn’t the proposed approach overfit to the

P10 Though the proposed approach is compared with baselines on a large number of datasets, analysis is missing. When the proposed approach is promising over the other approaches and when it is not?

**Strengths And Weaknesses:**

Strengths:

A simple (i.e., naive) hybrid approach to enhance the performance of online class imbalance learning.

Comparison on 30 datasets are performed.

Weaknesses:

The proposed approach requires a huge buffer to keep validation data for evaluating the performance. I looks like contradicting to the standard setting of online learning.

Related to the above point, it is not clear whether the comparison with the baseline approaches is fair. Do they also use such huge memory?

Though the performance is evaluated on 30 datasets, the analysis is missing. Therefore, when and why the proposed approach is promising/not promising over the other baselines are not clear.

---

> ### Author Response · Authors · 2024-08-04
> **Response to Reviewer gCJ4**
>
> We would like to thank the reviewer for the very detailed feedback and the constructive comments.
>
> *Weaknesses 1-2* Please refer to *Requested Changes 4*.
>
> *Weaknesses 3* Please refer to *Requested Changes 6*.
>
> *Requested Changes 1 on P3* We give a further illustration of this point in a separate subsection, 2.1.4, and explain why we want to optimize the performance metric directly.
>
> *Requested Changes 2 on P5* Thanks for pointing out this mistake. We have corrected formula (2) by formulating it as a constrained optimization problem. We want to emphasize that our model should be viewed as bi-level optimization since we're considering optimizing the two layers (classifier and cost vector) together and highlighting the novel bi-level architecture of our method.
>
> *Requested Changes 3 on P6* We apologize for the confusion here and have corrected it. The cost function remains untouched in this case. We want to express that the probability of minority class is upscaled, which can be quantified by the cost matrix.
>
> *Requested Changes 4 on P8*
>
> We acknowledge the extra storage requirement as a weakness in our current method compared to some existing methods that do not use any extra storage at all. We want first to point out one point: in literature, we usually consider the storage complexity, i.e., the size of past samples the algorithm has to store, to indicate the memory complexity. For example, in our case, the storage complexity is the buffer size $|\mathcal{B}|$ instead of the oversampled buffer size $|\mathcal{B}^\prime|$. This is because the past samples have to be held in the memory altogether, and the temporary synthesized samples can be processed one by one without storing everything in the memory, whose memory consumption is then negligible. Therefore, we compare the storage requirements between the methods used in this work.
>
> Second, certain storage requirements are generally acceptable in the literature, especially the data-level method (please refer to Section 2.1.1). Many methods are based on resampling and clustering; this necessitates storage, which is usually comparable to ours. For example, [1-4] all require certain storage in the form of a sliding window or chunk. We want to emphasize that despite the extra storage burden, it is constant and won't increase with the length of the stream. This means no matter whether the length of the stream is 5,000 or much higher, such as 500,000, the storage requirement is the same. This is practical in many real-world scenarios, such as online edge machine learning, where the stream can be infinitely large, but certain storage (while limited) is accessible. In practice, if additional storage is truly unavailable for our method, an adaptive generative model (such as online clustering) can be used to generate samples in replacement of the buffer. In this work, we focus on the current buffer scheme for simplicity.
>
> To make a fair comparison, we have taken AI-WSELM [1], one of the SOTA methods that requires extra storage, as our compared baseline. The extra storage of AI-WSELM was set to be $300$, being larger than our method. Other baselines, such as MOOB, MUOB, and BEDCOE, do not require extra storage. We have also added an additional comparison with Online SMOTE Bagging [4], which also requires extra storage. We set the size of the sliding window to $100$ for each class (i.e., at least $200$ samples to be stored for all classes), being comparable to ours. Our method outperforms Online SMOTE Bagging with a comparable storage budget and time cost. Please refer to Appendix F for the comparison.
>
> Besides, we have designed an ablation baseline named OECV-ea in the original submission that uses extra data, but no evolutionary algorithm is applied to demonstrate that OECV does not simply gain an advantage by extra storage. In the ablation study, OECV outperforms OECV-ea, meaning our method does not gain improvement solely from the extra data. Please refer to Section 3.4.1 for the definition of OECV-ea and Section 4.3 for the comparison. Appendix A includes a runtime comparison between our method and baselines.
>
> Finally, we do not use the buffered samples to learn the classifier for two reasons: 1) The online classifiers are better to update once the new sample is available, while the buffer contains old samples that have been used. 2) The buffer contains the past 200 samples which are local and may not include samples from all classes. Therefore, it is not suitable to train the classifier from the buffer directly.

---

> > ### Author Response · Authors · 2024-08-04
> > **Further Response to Reviewer gCJ4**
> >
> > *Requested Changes 5 on P9* In online learning literature, we do not consider the train-test split as in offline learning. Instead, we consider the test-then-train process, as we discussed in Section 3.1. The performance is evaluated along the stream after the pretraining stage using prequential evaluation [5-6].
> >
> > *Requested Changes 6 on P10*  We have added further analysis on this point using Spearman correlation. The key conclusion is that our method can improve performance on highly imbalanced datasets or datasets with few classes and may not be advantageous in other cases. Please refer to Section 4.2 for a detailed discussion.
> >
> > Thank you again for engaging with us and for your valuable feedback.
> >
> > [1] Jiongming Qin, Cong Wang, Qinhong Zou, Yubin Sun, and Bin Chen. Active learning with extreme learning machine for online imbalanced multiclass classification. Knowledge-Based Systems, 231:107385, 2021
> >
> > [2] Siqi Ren, Wen Zhu, Bo Liao, Zeng Li, Peng Wang, Keqin Li, Min Chen, and Zejun Li. Selection-based resampling ensemble algorithm for nonstationary imbalanced stream data learning. Knowledge-Based Systems, 163:705–722, 2019.
> >
> > [3] Alberto Cano and Bartosz Krawczyk. Rose: robust online self-adjusting ensemble for continual learning on imbalanced drifting data streams. Machine Learning, 111(7):2561–2599, 2022.
> >
> > [4] Wang, Boyu, and Joelle Pineau. "Online bagging and boosting for imbalanced data streams." *IEEE Transactions on Knowledge and Data Engineering* 28.12 (2016): 3353-3366.
> >
> > [5] Gama, Joao, Raquel Sebastiao, and Pedro Pereira Rodrigues. "On evaluating stream learning algorithms." *Machine learning* 90 (2013): 317-346.
> >
> > [6] Gama, Joao, Raquel Sebastiao, and Pedro Pereira Rodrigues. "Issues in evaluation of stream learning algorithms." *Proceedings of the 15th ACM SIGKDD international conference on Knowledge discovery and data mining*. 2009.

---

### Author Response · Authors · 2024-08-04
**Overall Response to Reviewers**

We deeply appreciate the valuable comments provided by the reviewers. We have tried our best to address the questions from the reviewers and revised our manuscript to the latest version. The reviewer’s suggestions significantly helped us to clarify our motivation and contribution. We have modified our paper according to the comments from all reviewers and highlighted the modified parts in **orange** for visibility. Please refer to the latest revised manuscript for details.

We summarized key modifications as follows. As to other questions, please refer to the response to each reviewer and revision of our manuscript.

1. We noticed that all reviewers are concerned about the hyperparameters in our method. We summarize four crucial hyperparameters in our method: updating frequency $f$,  population size $|\mathcal{V}|$, buffer size $|\mathcal{B}|$, and oversampling rate $r$. We have added a Remark2 on the influence of four hyperparameters at the end of Section 3.2, with corresponding experiments in Appendix E to verify the claims. In a nutshell, the influence of four hyperparameters is straightforward: the higher the $|\mathcal{V}|$, $|\mathcal{B}|$ and $r$, or a smaller $f$, will increase the performance while increasing the time consumption (for all four hyperparameters) and storage consumption (for $|\mathcal{B}|$). Besides, their values are relatively robust within a certain range in the sense that the same set of hyperparameters are used, which still makes OECV outperform baselines in a large number of datasets.
2. We noticed that two reviewers (reviewer gCJ4 and MURu) questioned the clarity of how the assumptions of existing hurt their performance. For a more detailed illustration of this point, we note that all three categories suffer from these issues. Therefore, we move the discussion from the end of subsection 2.1.1 to a separate subsection 2.1.4.
3. We also highlight that most data-level methods (as in Section 2.1.1 ) require the storage of extra data. This shows that storing samples is generally acceptable in literature, as what we do in this work. We added a comparison between our method and Online SMOTE Bagging, one of the SOTA data-level methods that uses the same extra storage as ours, to Appendix F. This supplies a fair comparison in addition to the experiments in the main paper.
4. Besides the running time comparison in Appendix A of our first submission, we also include a computational complexity analysis (both storage complexity and time complexity) in Section 3.4.4 from a theoretical perspective.
5. We apologize for the typo of one hyperparameter setting: we corrected the original experimental setting that the oversampling rate $r$ should be $5$ instead of $3$. The original experimental results were all correct.

---

### Decision · Action_Editor_Xe8d · 2024-09-08

**Recommendation:** Accept with minor revision

**Comment:**

The paper presents a novel approach to the class imbalance problem in online learning environments by introducing the Online Evolutionary Cost Vector (OECV). This method transforms the class imbalance challenge into a bi-level optimization problem, which combines an online classifier with a dynamic evolutionary algorithm to adaptively learn a cost vector. The approach is validated on 30 datasets, showing promising results against four state-of-the-art (SOTA) baselines.

Overall, the paper makes a valuable contribution to the field of online learning with class imbalance, particularly by incorporating evolutionary algorithms into the process. While the original paper raised several concerns, particularly around hyperparameter sensitivity, computational costs, overfitting, the authors have made significant efforts to address these issues raised by the reviewers.

However, one reviewer still questions the large buffer size requirement, so it would be helpful for the authors to discuss this issue further in the final version.

**Audience:**

Yes

**Claims And Evidence:**

Yes